# DiffEnc: Variational Diffusion with a Learned Encoder

**Beatrix M. G. Nielsen,**[*1] **Anders Christensen,**[1,2] **Andrea Dittadi,**[†2,4] **Ole Winther**[†1,3,5]

[1]Technical University of Denmark, [2]Helmholtz AI, Munich, [3]University of Copenhagen,
[4]Max Planck Institute for Intelligent Systems, [5]Copenhagen University Hospital

## Abstract

Diffusion models may be viewed as hierarchical variational autoencoders (VAEs) with two improvements: *parameter sharing* for the conditional distributions in the generative process and *efficient* computation of the loss as independent terms over the hierarchy. We consider two changes to the diffusion model that retain these advantages while adding flexibility to the model. Firstly, we introduce a data- and depth-dependent mean function in the diffusion process, which leads to a modified diffusion loss. Our proposed framework, DiffEnc, achieves a statistically significant improvement in likelihood on CIFAR-10. Secondly, we let the ratio of the noise variance of the reverse encoder process and the generative process be a free *weight parameter* rather than being fixed to 1. This leads to theoretical insights: For a finite depth hierarchy, the evidence lower bound (ELBO) can be used as an objective for a weighted diffusion loss approach and for optimizing the noise schedule specifically for inference. For the infinite-depth hierarchy, on the other hand, the weight parameter has to be 1 to have a well-defined ELBO.

## 1 Introduction

Diffusion models (Sohl-Dickstein et al., 2015; Song & Ermon, 2019; Ho et al., 2020) are versatile generative models that have risen to prominence in recent years thanks to their state-of-the-art performance in the generation of images (Dhariwal & Nichol, 2021; Karras et al., 2022), video (Ho et al., 2022; Höppe et al., 2022; Harvey et al., 2022), speech (Kong et al., 2020; Jeong et al., 2021; Chen et al., 2020), and music (Huang et al., 2023; Schneider et al., 2023). In particular, in image generation, diffusion models are state of the art both in terms of visual quality (Karras et al., 2022; Kim et al., 2022a; Zheng et al., 2022; Hoogeboom et al., 2023; Kingma & Gua, 2023; Lou & Ermon, 2023) and density estimation (Kingma et al., 2021; Nichol & Dhariwal, 2021; Song et al., 2021).

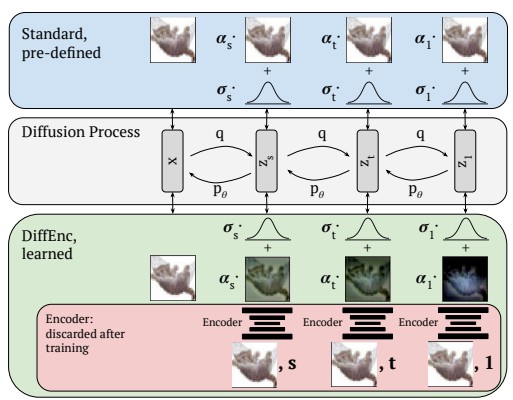

Figure 1: Overview of DiffEnc compared to standard diffusion models. The effect of the encoding has been amplified 5x for the sake of illustration.

Diffusion models can be seen as a time-indexed hierarchy over latent variables generated sequentially, conditioning only on the latent vector from the previous step. As such, diffusion models can be understood as hierarchical variational autoencoders (VAEs) (Kingma & Welling, 2013; Rezende et al., 2014; Sønderby et al., 2016) with three restrictions: (1) the *forward diffusion process*—the *inference model* in variational inference—is fixed and remarkably simple; (2) the *generative model is Markovian*—each (time-indexed) layer of latent variables is generated conditioning only on the previous layer; (3) *parameter sharing*—all steps of the generative model share the same parameters.

---

[*]Correspondence to: <bmgi@dtu.dk>.
[†]Equal advising.

The simplicity of the forward process (1) and the Markov property of the generative model (2) allow the evidence lower bound (ELBO) to be expressed as an expectation over the layers of random variables, i.e., an expectation over time from the stochastic process perspective. Thanks to the heavy parameter sharing in the generative model (3), this expectation can be estimated effectively with a single Monte Carlo sample. These properties make diffusion models highly scalable and flexible, despite the constraints discussed above.

*In this work, we relax assumption (1) to improve the flexibility of diffusion models while retaining their scalability.* Specifically, we shift away from assuming a constant diffusion process, while still maintaining sufficient simplicity to express the ELBO as an expectation over time. We introduce a *time-dependent encoder* that parameterizes the mean of the diffusion process: instead of the original image $\mathbf{x}$, the learned denoising model is tasked with predicting $\mathbf{x}_t$, which is the encoded image at time $t$. Crucially, this encoder is exclusively employed during the training phase and not utilized during the sampling process. As a result, the proposed class of diffusion models, *DiffEnc*,

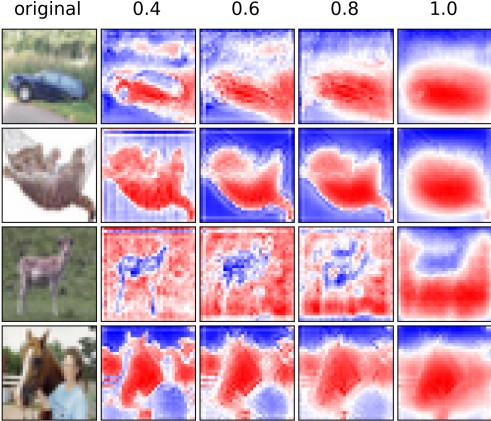

Figure 2: Changes induced by the encoder on the encoded image at different timesteps: $(\mathbf{x}_t - \mathbf{x}_s)/(t-s)$ for $t = 0.4, 0.6, 0.8, 1.0$ and $s = t - 0.1$. Changes have been summed over the channels with red and blue denoting positive and negative changes, respectively. For $t \rightarrow 1$, global properties such as approximate position of objects are encoded, where for smaller $t$ changes are more fine-grained and tend to enhance high-contrast within objects and/or between object and background.

is more flexible than standard diffusion models without affecting sampling time. To arrive at the negative log likelihood loss for DiffEnc, Eq. (18), we will first show how we can introduce a time-dependent encoder to the diffusion process and how this introduces an extra term in the loss if we use the usual expression for the mean in the generative model, Section 3. We then show how we can counter this extra term, using a certain parametrization of the encoder, Section 4.

We conduct experiments on MNIST, CIFAR-10 and ImageNet32 with two different parameterizations of the encoder and find that, with a trainable encoder, DiffEnc improves total likelihood on CIFAR-10 and improves the latent loss on all datasets without damaging the diffusion loss. We observe that the changes to $\mathbf{x}_t$ are significantly different for early and late timesteps, demonstrating the non-trivial, time-dependent behavior of the encoder (see Fig. 2).

In addition, we investigate the relaxation of a common assumption in diffusion models: That the variance of the generative process, $\sigma_P^2$, is equal to the variance of the reverse formulation of the forward diffusion process, $\sigma_Q^2$. This introduces an additional term in the diffusion loss, which can be interpreted as a weighted loss (with time-dependent weights $w_t$). We then analytically derive the optimal $\sigma_P^2$. While this is relevant when training in discrete time (i.e., with a finite number of layers) or when sampling, we prove that the ELBO is maximized in the continuous-time limit when the variances are equal (in fact, the ELBO diverges if the variances are not equal).

Our main contributions can be summarized as follows:

- We define a new, more powerful class of diffusion models—named *DiffEnc*—by introducing a time-dependent encoder in the diffusion process. This encoder improves the flexibility of diffusion models but does not affect sampling time, as it is only needed during training.

- We analyse the assumption of forward and backward variances being equal, and prove that (1) by relaxing this assumption, the diffusion loss can be interpreted as a weighted loss, and (2) in continuous time, the optimal ELBO is achieved when the variances are equal— in fact, if the variances are not equal in continuous time, the ELBO is not well-defined.

- We perform extensive density estimation experiments and show that DiffEnc achieves a statistically significant improvement in likelihood on CIFAR-10.

The paper is organized as follows: In Section 2 we introduce the notation and framework from Variational Diffusion Models (VDM; Kingma et al., 2021); in Section 3 we derive the general formulation

of DiffEnc by introducing a depth-dependent encoder; in Section 4 we introduce the encoder parameterizations used in our experiments and modify the generative model to account for the change in the diffusion loss due to the encoder; in Section 5 we present our experimental results.

## 2 PRELIMINARIES ON VARIATIONAL DIFFUSION MODELS

We begin by introducing the VDM formulation (Kingma et al., 2021) of diffusion models. We define a hierarchical generative model with $T + 1$ layers of latent variables:

$$p_{\boldsymbol{\theta}}(\mathbf{x}, \mathbf{z}) = p(\mathbf{x}|\mathbf{z}_0)p(\mathbf{z}_1) \prod_{i=1}^{T} p_{\boldsymbol{\theta}}(\mathbf{z}_{s(i)}|\mathbf{z}_{t(i)}) \tag{1}$$

with $\mathbf{x} \in \mathcal{X}$ a data point, $\boldsymbol{\theta}$ the model parameters, $s(i) = \frac{i-1}{T}$, $t(i) = \frac{i}{T}$, and $p(\mathbf{z}_1) = \mathcal{N}(\mathbf{0}, \mathbf{I})$. In the following, we will drop the index $i$ and assume $0 \leq s < t \leq 1$. We define a diffusion process $q$ with marginal distribution:

$$q(\mathbf{z}_t|\mathbf{x}) = \mathcal{N}(\alpha_t\mathbf{x}, \sigma_t^2\mathbf{I}) \tag{2}$$

where $t \in [0, 1]$ is the time index and $\alpha_t, \sigma_t$ are positive scalar functions of $t$. Requiring Eq. (2) to hold for any $s$ and $t$, the conditionals turns out to be:

$$q(\mathbf{z}_t|\mathbf{z}_s) = \mathcal{N}(\alpha_{t|s}\mathbf{z}_s, \sigma_{t|s}^2\mathbf{I}) ,$$

where

$$\alpha_{t|s} = \frac{\alpha_t}{\alpha_s} , \ \sigma_{t|s}^2 = \sigma_t^2 - \alpha_{t|s}^2\sigma_s^2 .$$

Using Bayes' rule, we can reverse the direction of the diffusion process:

$$q(\mathbf{z}_s|\mathbf{z}_t, \mathbf{x}) = \mathcal{N}(\boldsymbol{\mu}_Q, \sigma_Q^2\mathbf{I}) \tag{3}$$

with

$$\sigma_Q^2 = \frac{\sigma_{t|s}^2\sigma_s^2}{\sigma_t^2}, \ \boldsymbol{\mu}_Q = \frac{\alpha_{t|s}\sigma_s^2}{\sigma_t^2}\mathbf{z}_t + \frac{\alpha_s\sigma_{t|s}^2}{\sigma_t^2}\mathbf{x} . \tag{4}$$

We can now express the diffusion process in a way that mirrors the generative model in Eq. (1):

$$q(\mathbf{z}|\mathbf{x}) = q(\mathbf{z}_1|\mathbf{x}) \prod_{i=1}^{T} q(\mathbf{z}_{s(i)}|\mathbf{z}_{t(i)}, \mathbf{x}) \tag{5}$$

and we can define one step of the generative process in the same functional form as Eq. (3):

$$p_{\boldsymbol{\theta}}(\mathbf{z}_s|\mathbf{z}_t) = \mathcal{N}(\boldsymbol{\mu}_P, \sigma_P^2\mathbf{I})$$

with

$$\boldsymbol{\mu}_P = \frac{\alpha_{t|s}\sigma_s^2}{\sigma_t^2}\mathbf{z}_t + \frac{\alpha_s\sigma_{t|s}^2}{\sigma_t^2}\hat{\mathbf{x}}_{\boldsymbol{\theta}}(\mathbf{z}_t, t) , \tag{6}$$

where $\hat{\mathbf{x}}_{\boldsymbol{\theta}}$ is a learned model with parameters $\boldsymbol{\theta}$. In a diffusion model, the denoising variance $\sigma_P^2$ is usually chosen to be equal to the reverse diffusion process variance: $\sigma_P^2 = \sigma_Q^2$. While initially we do not make this assumption, we will prove this to be optimal in the continuous-time limit. Following VDM, we parameterize the noise schedule through the signal-to-noise ratio (SNR):

$$\text{SNR}(t) \equiv \frac{\alpha_t^2}{\sigma_t^2}$$

and its logarithm: $\lambda_t \equiv \log \text{SNR}(t)$. We will use the variance-preserving formulation in all our experiments: $\alpha_t^2 = 1 - \sigma_t^2 = \text{sigmoid}(\lambda_t)$.

The evidence lower bound (ELBO) of the model defined above is:

$$\log p_{\boldsymbol{\theta}}(\mathbf{x}) \geq \mathbb{E}_{q(\mathbf{z}|\mathbf{x})}\left[\frac{p_{\boldsymbol{\theta}}(\mathbf{x}|\mathbf{z})p_{\boldsymbol{\theta}}(\mathbf{z})}{q(\mathbf{z}|\mathbf{x})}\right] \equiv \text{ELBO}(\mathbf{x})$$

The loss $\mathcal{L} \equiv -\text{ELBO}$ is the sum of a reconstruction ($\mathcal{L}_0$), diffusion ($\mathcal{L}_T$), and latent ($\mathcal{L}_1$) loss:

$$\mathcal{L} = \mathcal{L}_0 + \mathcal{L}_T + \mathcal{L}_1$$
$$\mathcal{L}_0 = -\mathbb{E}_{q(\mathbf{z}_0|\mathbf{x})}\left[\log p(\mathbf{x}|\mathbf{z}_0)\right]$$
$$\mathcal{L}_1 = D_{\text{KL}}(q(\mathbf{z}_1|\mathbf{x})\|p(\mathbf{z}_1))\,,$$

where the expressions for $\mathcal{L}_0$ and $\mathcal{L}_1$ are derived in Appendix D. Thanks to the matching factorization of the generative and reverse noise processes—see Eqs. (1) and (5)—and the availability of $q(\mathbf{z}_t|\mathbf{x})$ in closed form because $q$ is Markov and Gaussian, the diffusion loss $\mathcal{L}_T$ can be written as a sum or as an expectation over the layers of random variables:

$$\mathcal{L}_T(\mathbf{x}) = \sum_{i=1}^{T} \mathbb{E}_{q(\mathbf{z}_{t(i)}|\mathbf{x})}\left[D_{\text{KL}}(q(\mathbf{z}_{s(i)}|\mathbf{z}_{t(i)},\mathbf{x})\|p_{\boldsymbol{\theta}}(\mathbf{z}_{s(i)}|\mathbf{z}_{t(i)}))\right] \tag{7}$$

$$= T\,\mathbb{E}_{i \sim U\{1,T\},q(\mathbf{z}_{t(i)}|\mathbf{x})}\left[D_{\text{KL}}(q(\mathbf{z}_{s(i)}|\mathbf{z}_{t(i)},\mathbf{x})\|p_{\boldsymbol{\theta}}(\mathbf{z}_{s(i)}|\mathbf{z}_{t(i)}))\right]\,, \tag{8}$$

where $U\{1,T\}$ is the uniform distribution over the indices 1 through $T$. Since all distributions are Gaussian, the KL divergence has a closed-form expression (see Appendix E):

$$D_{\text{KL}}(q(\mathbf{z}_s|\mathbf{z}_t,\mathbf{x})\|p_{\boldsymbol{\theta}}(\mathbf{z}_s|\mathbf{z}_t)) = \frac{d}{2}\left(w_t - 1 - \log w_t\right) + \frac{w_t}{2\sigma_Q^2}\|\boldsymbol{\mu}_P - \boldsymbol{\mu}_Q\|_2^2\,, \tag{9}$$

where the green part is the difference from using $\sigma_P^2 \neq \sigma_Q^2$ instead of $\sigma_P^2 = \sigma_Q^2$, and we have defined the weighting function

$$w_t = \frac{\sigma_{Q,t}^2}{\sigma_{P,t}^2}$$

and the dependency of $\sigma_{Q,t}^2$ and $\sigma_{P,t}^2$ on $s$ is left implicit, since the step size $t - s = \frac{1}{T}$ is fixed. The optimal generative variance can be computed in closed-form (see Appendix F):

$$\sigma_P^2 = \sigma_Q^2 + \frac{1}{d}\mathbb{E}_{q(\mathbf{x},\mathbf{z}_t)}\left[\|\boldsymbol{\mu}_P - \boldsymbol{\mu}_Q\|_2^2\right]\,.$$

## 3 DIFFENC

The main component of DiffEnc is the time-dependent encoder, which we will define as $\mathbf{x}_t \equiv \mathbf{x}_{\boldsymbol{\phi}}(\lambda_t)$, where $\mathbf{x}_{\boldsymbol{\phi}}(\lambda_t)$ is some function with parameters $\boldsymbol{\phi}$ dependent on $\mathbf{x}$ and $t$ through $\lambda_t \equiv \log \text{SNR}(t)$. The generalized version of Eq. (2) is then:

$$q(\mathbf{z}_t|\mathbf{x}) = \mathcal{N}(\alpha_t \mathbf{x}_t, \sigma_t^2 \mathbf{I})\,. \tag{10}$$

Fig. 1 visualizes this change to the diffusion process, and a diagram is provided in Appendix A. Requiring that the process is consistent upon marginalization, i.e., $q(\mathbf{z}_t|\mathbf{x}) = \int q(\mathbf{z}_t|\mathbf{z}_s,\mathbf{x})q(\mathbf{z}_s|\mathbf{x})d\mathbf{z}_s$, leads to the following conditional distributions (see Appendix B):

$$q(\mathbf{z}_t|\mathbf{z}_s,\mathbf{x}) = \mathcal{N}(\alpha_{t|s}\mathbf{z}_s + \alpha_t(\mathbf{x}_t - \mathbf{x}_s), \sigma_{t|s}^2 \mathbf{I})\,, \tag{11}$$

where an additional mean shift term is introduced by the depth-dependent encoder. As in Section 2, we can derive the reverse process (see Appendix C):

$$q(\mathbf{z}_s|\mathbf{z}_t,\mathbf{x}) = \mathcal{N}(\boldsymbol{\mu}_Q, \sigma_Q^2 \mathbf{I}) \tag{12}$$

$$\boldsymbol{\mu}_Q = \frac{\alpha_{t|s}\sigma_s^2}{\sigma_t^2}\mathbf{z}_t + \frac{\alpha_s\sigma_{t|s}^2}{\sigma_t^2}\mathbf{x}_t + \alpha_s(\mathbf{x}_s - \mathbf{x}_t) \tag{13}$$

with $\sigma_Q^2$ given by Eq. (4). We show how we parameterize the encoder in Section 4.

**Infinite-depth limit.** Kingma et al. (2021) derived the continuous-time limit of the diffusion loss, that is, the loss in the limit of $T \rightarrow \infty$. We can extend that result to our case. Using $\boldsymbol{\mu}_Q$ from Eq. (13) and $\boldsymbol{\mu}_P$ from Eq. (6), the KL divergence in the unweighted case, i.e., $\frac{1}{2\sigma_Q^2}\|\boldsymbol{\mu}_P - \boldsymbol{\mu}_Q\|_2^2$, can be rewritten in the following way, as shown in Appendix G:

$$\frac{1}{2\sigma_Q^2}\|\boldsymbol{\mu}_P - \boldsymbol{\mu}_Q\|_2^2 = -\frac{1}{2}\Delta\text{SNR}\left\|\hat{\mathbf{x}}_{\boldsymbol{\theta}}(\mathbf{z}_t,t) - \mathbf{x}_{\boldsymbol{\phi}}(\lambda_t) - \text{SNR}(s)\frac{\Delta\mathbf{x}}{\Delta\text{SNR}}\right\|_2^2\,,$$

where $\Delta\mathbf{x} \equiv \mathbf{x}_\phi(\lambda_t) - \mathbf{x}_\phi(\lambda_s)$ and similarly for the SNR. In Appendix G, we also show that, as $T \to \infty$, the expression for the optimal $\sigma_P$ tends to $\sigma_Q$ and the additional term in the diffusion loss arising from allowing $\sigma_P^2 \neq \sigma_Q^2$ tends to 0. This result is in accordance with prior work on variational approaches to stochastic processes (Archambeau et al., 2007). We have shown that, in the continuous limit, the ELBO has to be an unweighted loss (in the sense that $w_t = 1$). In the remainder of the paper, we will use the continuous formulation and thus set $w_t = 1$. It is of interest to consider optimized weighted losses for a finite number of layers, however, we leave this for future research.

The infinite-depth limit of the diffusion loss, $\mathcal{L}_\infty(\mathbf{x}) \equiv \lim_{T\to\infty} \mathcal{L}_T(\mathbf{x})$, becomes (Appendix G):

$$\mathcal{L}_\infty(\mathbf{x}) = -\frac{1}{2}\mathbb{E}_{t\sim U(0,1)}\mathbb{E}_{q(\mathbf{z}_t|\mathbf{x})}\left[\frac{d\text{SNR}(t)}{dt}\left\|\hat{\mathbf{x}}_{\boldsymbol{\theta}}(\mathbf{z}_t,t) - \mathbf{x}_\phi(\lambda_t) - \frac{d\mathbf{x}_\phi(\lambda_t)}{d\lambda_t}\right\|_2^2\right] . \tag{14}$$

$\mathcal{L}_\infty(\mathbf{x})$ thus is very similar to the standard continuous-time diffusion loss from VDM, though with an additional gradient stemming from the mean shift term. In Section 4, we will develop a modified generative model to counter this extra term. In Appendix H, we derive the stochastic differential equation (SDE) describing the generative model of DiffEnc in the infinite-depth limit.

## 4 PARAMETERIZATION OF THE ENCODER AND GENERATIVE MODEL

We now turn to the parameterization of the encoder $\mathbf{x}_\phi(\lambda_t)$. The reconstruction and latent losses impose constraints on how the encoder should behave at the two ends of the hierarchy of latent variables: The likelihood we use is constructed such that the reconstruction loss, derived in Appendix D, is minimized when $\mathbf{x}_\phi(\lambda_0) = \mathbf{x}$. Likewise, the latent loss is minimized by $\mathbf{x}_\phi(\lambda_1) = \mathbf{0}$. In between, for $0 < t < 1$, a non-trivial encoder can improve the diffusion loss.

We propose two related parameterizations of the encoder: a trainable one, which we will denote by $\mathbf{x}_\phi$, and a simpler, non-trainable one, $\mathbf{x}_{\text{nt}}$, where $nt$ stands for non-trainable. Let $\mathbf{y}_\phi(\mathbf{x}, \lambda_t)$ be a neural network with parameters $\phi$, denoted $\mathbf{y}_\phi(\lambda_t)$ for brevity. We define the trainable encoder as

$$\mathbf{x}_\phi(\lambda_t) = (1 - \sigma_t^2)\mathbf{x} + \sigma_t^2\mathbf{y}_\phi(\lambda_t) = \alpha_t^2\mathbf{x} + \sigma_t^2\mathbf{y}_\phi(\lambda_t) \tag{15}$$

and the non-trainable encoder as

$$\mathbf{x}_{\text{nt}}(\lambda_t) = \alpha_t^2\mathbf{x} . \tag{16}$$

More motivation for these parameterizations can be found in Appendix I. The trainable encoder $\mathbf{x}_\phi$ is initialized with $\mathbf{y}_\phi(\lambda_t) = 0$, so at the start of training it acts as the non-trainable encoder $\mathbf{x}_{\text{nt}}$ (but differently from the VDM, which corresponds to the identity encoder).

To better fit the infinite-depth diffusion loss in Eq. (14), we define a new mean, $\boldsymbol{\mu}_P$, of the generative model $p_{\boldsymbol{\theta}}(\mathbf{z}_s|\mathbf{z}_t)$ which is a modification of Eq. (6). Concretely, we would like to introduce a counterterm in $\boldsymbol{\mu}_P$ that, when taking the continuous-limit, approximately counters $\frac{d\mathbf{x}_\phi(\lambda_t)}{d\lambda_t}$. This term should be expressed in terms of $\hat{\mathbf{x}}_{\boldsymbol{\theta}}(\lambda_t)$ rather than $\mathbf{x}_\phi$. For the non-trainable encoder, we have

$$\frac{d\mathbf{x}_{\text{nt}}(\lambda_t)}{d\lambda_t} = \alpha_t^2\sigma_t^2\mathbf{x} = \sigma_t^2\mathbf{x}_{\text{nt}}(\lambda_t) .$$

Therefore, for the non-trainable encoder, we can use $\sigma_t^2\hat{\mathbf{x}}_{\boldsymbol{\theta}}(\lambda_t)$ as an approximation of $\frac{d\mathbf{x}_{\text{nt}}(\lambda_t)}{d\lambda_t}$. The trainable encoder is more complicated because it also contains the derivative of $\mathbf{y}_\phi$ that we cannot as straightforwardly express in terms of $\hat{\mathbf{x}}_{\boldsymbol{\theta}}$. We therefore choose to approximate $\frac{d\mathbf{x}_\phi(\lambda_t)}{d\lambda_t}$ the same way as $\frac{d\mathbf{x}_{\text{nt}}(\lambda_t)}{d\lambda_t}$. We leave it for future work to explore different strategies for approximating this gradient. Since we use the same approximation for both encoders, in the following we will write $\mathbf{x}_\phi(\lambda_t)$ for both.

With the chosen counterterm, which in the continuous limit should approximately cancel out the effect of the mean shift term in Eq. (13), the new mean, $\boldsymbol{\mu}_P$, is defined as:

$$\boldsymbol{\mu}_P = \frac{\alpha_{t|s}\sigma_s^2}{\sigma_t^2}\mathbf{z}_t + \frac{\alpha_s\sigma_{t|s}^2}{\sigma_t^2}\hat{\mathbf{x}}_{\boldsymbol{\theta}}(\lambda_t) + \alpha_s(\lambda_s - \lambda_t)\sigma_t^2\hat{\mathbf{x}}_{\boldsymbol{\theta}}(\lambda_t) \tag{17}$$

Table 1: Comparison of average bits per dimension (BPD) over 3 seeds on CIFAR-10 and ImageNet32 with other work. Types of models are Continuous Flow (Flow), Variational Auto Encoders (VAE), AutoRegressive models (AR) and Diffusion models (Diff). We only compare with results achieved without data augmentation. DiffEnc with a trainable encoder improves performance of the VDM on CIFAR-10. Results on ImageNet marked with $*$ are on the (Van Den Oord et al., 2016) version of ImageNet which is no longer officially available. Results without $*$ are on the (Chrabaszcz et al., 2017) version of ImageNet, which is from the official ImageNet website. Results from (Zheng et al., 2023) are without importance sampling, since importance sampling could also be added to our approach.

| Model | Type | CIFAR-10 | ImageNet 32×32 |
|---|---|---|---|
| Flow Matching OT (Lipman et al., 2022) | Flow | 2.99 | 3.53 |
| Stochastic Int. (Albergo & Vanden-Eijnden, 2022) | Flow | 2.99 | 3.48* |
| NVAE (Vahdat & Kautz, 2020) | VAE | 2.91 | 3.92* |
| Image Transformer (Parmar et al., 2018) | AR | 2.90 | 3.77* |
| VDVAE (Child, 2020) | VAE | 2.87 | 3.80* |
| ScoreFlow (Song et al., 2021) | Diff | 2.83 | 3.76* |
| Sparse Transformer (Child et al., 2019) | AR | 2.80 | − |
| Reflected Diffusion Models (Lou & Ermon, 2023) | Diff | 2.68 | 3.74* |
| VDM (Kingma et al., 2021) *(10M steps)* | Diff | 2.65 | 3.72* |
| ARDM (Hoogeboom et al., 2021) | AR | 2.64 | − |
| Flow Matching TN (Zheng et al., 2023) | Flow | 2.60 | 3.45 |
| *Our experiments (8M and 1.5M steps, 3 seed avg)* | | | |
| VDM with **v**-parameterization | Diff | 2.64 | 3.46 |
| DiffEnc Trainable (ours) | Diff | 2.62 | 3.46 |

Similarly to above, we derive the infinite-depth diffusion loss when the encoder is parameterized by Eq. (15) by taking the limit of $\mathcal{L}_T$ for $T \to \infty$ (see Appendix J):

$$\mathcal{L}_\infty(\mathbf{x}) = -\frac{1}{2}\mathbb{E}_{\boldsymbol{\epsilon}, t \sim U[0,1]}\left[\frac{de^{\lambda_t}}{dt}\left\|\hat{\mathbf{x}}_{\boldsymbol{\theta}}(\mathbf{z}_t, \lambda_t) + \sigma_t^2 \hat{\mathbf{x}}_{\boldsymbol{\theta}}(\mathbf{z}_t, \lambda_t) - \mathbf{x}_{\boldsymbol{\phi}}(\lambda_t) - \frac{d\mathbf{x}_{\boldsymbol{\phi}}(\lambda_t)}{d\lambda_t}\right\|_2^2\right], \quad (18)$$

where $\mathbf{z}_t = \alpha_t \mathbf{x}_t + \sigma_t \boldsymbol{\epsilon}$ with $\boldsymbol{\epsilon} \sim \mathcal{N}(\mathbf{0}, \mathbf{I})$.

**v-parameterization.** In our experiments we use the **v**-prediction parameterization (Salimans & Ho, 2022) for our loss, which means that for the trainable encoder we use the loss

$$\mathcal{L}_\infty(\mathbf{x}) = -\frac{1}{2}\mathbb{E}_{\boldsymbol{\epsilon}, t \sim U[0,1]}\left[\lambda_t' \alpha_t^2 \left\|\mathbf{v}_t - \hat{\mathbf{v}}_{\boldsymbol{\theta}} + \sigma_t\left(\hat{\mathbf{x}}_{\boldsymbol{\theta}}(\lambda_t) - \mathbf{x}_{\boldsymbol{\phi}}(\lambda_t) + \mathbf{y}_{\boldsymbol{\phi}}(\lambda_t) - \frac{d\mathbf{y}_{\boldsymbol{\phi}}(\lambda_t)}{d\lambda_t}\right)\right\|_2^2\right]$$
$$(19)$$

and for the non-trainable encoder, we use

$$\mathcal{L}_\infty(\mathbf{x}) = -\frac{1}{2}\mathbb{E}_{\boldsymbol{\epsilon}, t \sim U[0,1]}\left[\lambda_t' \alpha_t^2 \left\|\mathbf{v}_t - \hat{\mathbf{v}}_{\boldsymbol{\theta}} + \sigma_t\left(\hat{\mathbf{x}}_{\boldsymbol{\theta}}(\lambda_t) - \mathbf{x}_{\boldsymbol{\phi}}(\lambda_t)\right)\right\|_2^2\right]. \quad (20)$$

Derivations of Eqs. (19) and (20) are in Appendix K. We note that when using the **v**-parametrization, as $t$ tends to 0, the loss becomes the same as for the $\boldsymbol{\epsilon}$-prediction parameterization. On the other hand, when $t$ tends to 1, the loss has a different behavior depending on the encoder: For the trainable encoder, we have that $\hat{\mathbf{v}}_{\boldsymbol{\theta}} \approx \frac{d\mathbf{y}_{\boldsymbol{\phi}}(\lambda_t)}{d\lambda_t}$, suggesting that the encoder can in principle guide the diffusion model. See Appendix L for a more detailed discussion.

## 5 EXPERIMENTS

In this section, we present our experimental setup and discuss the results.

Table 2: Comparison of the different components of the loss for DiffEnc-32-4 and VDMv-32 with fixed noise schedule on CIFAR-10. All quantities are in bits per dimension (BPD) with standard error over 3 seeds, and models are trained for 8M steps.

| Model | Total | Latent | Diffusion | Reconstruction |
|-------|-------|--------|-----------|----------------|
| VDMv-32 | $2.641 \pm 0.003$ | $0.0012 \pm 0.0$ | $2.629 \pm 0.003$ | $\underline{0.01} \pm (4 \times 10^{-6})$ |
| DiffEnc-32-4 | $\underline{2.620} \pm 0.006$ | $\underline{0.0007} \pm (3 \times 10^{-6})$ | $\underline{2.609} \pm 0.006$ | $\underline{0.01} \pm (4 \times 10^{-6})$ |

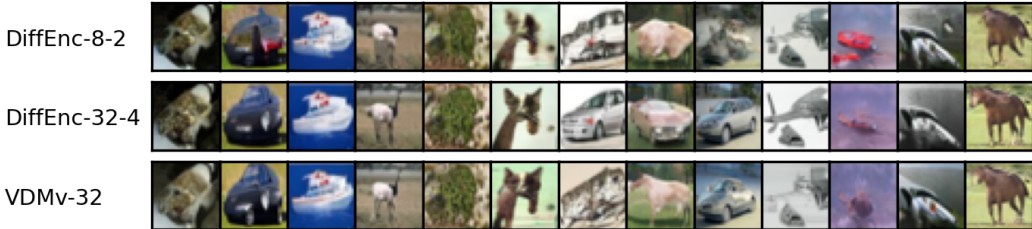

Figure 3: Comparison of unconditional samples of models. The small model struggles to make realistic images, while the large models are significantly better, as expected. For some images, details differ between the two large models, for others they disagree on the main element of the image. An example where the models make two different cars in column 9. An example where DiffEnc-32-4 makes a car and VDMv-32 makes a frog in column 7.

**Experimental Setup.** We evaluated variants of DiffEnc against a standard VDM baseline on MNIST (LeCun et al., 1998), CIFAR-10 (Krizhevsky et al., 2009) and ImageNet32 (Chrabaszcz et al., 2017). The learned prediction function is implemented as a U-Net (Ronneberger et al., 2015) consisting of convolutional ResNet blocks without any downsampling, following VDM (Kingma et al., 2021). The trainable encoder in DiffEnc is implemented with the same overall U-Net architecture, but with downsampling to resolutions 16x16 and 8x8. We will denote the models trained in our experiments by VDMv-$n$, DiffEnc-$n$-$m$, and DiffEnc-$n$-nt, where: VDMv is a VDM model with **v** parameterization, $n$ and $m$ are the number of ResNet blocks in the "downsampling" part of the **v**-prediction U-Net and of the encoder U-Net respectively, and *nt* indicates a non-trainable encoder for DiffEnc. On MNIST and CIFAR-10, we trained VDMv-8, DiffEnc-8-2, and DiffEnc-8-nt models. On CIFAR-10 we also trained DiffEnc-8-4, VDMv-32 and DiffEnc-32-4. On ImageNet32, we trained VDMv-32 and DiffEnc-32-8.

We used a linear log SNR noise schedule: $\lambda_t = \lambda_{max} - (\lambda_{max} - \lambda_{min}) \cdot t$. For the large models (VDMv-32, DiffEnc-32-4 and DiffEnc-32-8), we fixed the endpoints, $\lambda_{max}$ and $\lambda_{min}$, to the ones Kingma et al. (2021) found were optimal. For the small models (VDMv-8, DiffEnc-8-2 and DiffEnc-8-nt), we also experimented with learning the SNR endpoints. We trained all our models with either 3 or 5 seeds depending on the computational cost of the experiments. See more details on model structure and training in Appendix Q and on datasets in Appendix R.

**Results.** As we see in Table 1 the DiffEnc-32-4 model achieves a lower BPD score than previous non-flow work and the VDMv-32 on CIFAR-10. Since we do not use the encoder when sampling, this result means that the encoder is useful for learning a better generative model—with higher likelihoods—while sampling time is not adversely affected. We also see that VDMv-32 after 8M steps achieved a better likelihood bound, $2.64$ BPD, than the result reported by Kingma et al. (2021) for the $\epsilon$-parameterization after 10M steps, $2.65$ BPD. Thus, the **v**-parameterization gives an improved likelihood compared to $\epsilon$-parameterization. Table 2 shows that the difference in the total loss comes mainly from the improvement in diffusion loss for DiffEnc-32-4, which points to the encoder being helpful in the diffusion process. We provide Fig. 2, since it can be difficult to see what the encoder is doing directly from the encodings. From the heatmaps, we see that the encoder has learnt to do something different from how it was initialised and that it acts differently over $t$, making finer changes in earlier timesteps and more global changes in later timesteps. See Appendix W for more details. We note that the improvement in total loss is significant, since we get a p-value of $0.03$ for

Table 3: Comparison of the different components of the loss for DiffEnc-8-2, DiffEnc-8-nt and VDMv-8 on CIFAR-10. All quantities are in bits per dimension (BPD), with standard error, 5 seeds, 2M steps. Noise schedules are either fixed or with trainable endpoints.

| Model | Noise | Total | Latent | Diffusion | Reconstruction |
|-------|-------|-------|--------|-----------|----------------|
| VDMv-8 | fixed | $2.783 \pm 0.004$ | $0.0012 \pm 0.0$ | $2.772 \pm 0.004$ | $0.010 \pm (2 \times 10^{-5})$ |
| | trainable | $2.776 \pm 0.0006$ | $0.0033 \pm (2 \times 10^{-5})$ | $2.770 \pm 0.0006$ | $0.003 \pm (5 \times 10^{-5})$ |
| DiffEnc-8-2 | fixed | $2.783 \pm 0.004$ | $0.0006 \pm (3 \times 10^{-5})$ | $2.772 \pm 0.004$ | $0.010 \pm (3 \times 10^{-6})$ |
| | trainable | $2.783 \pm 0.003$ | $0.0034 \pm (2 \times 10^{-5})$ | $2.777 \pm 0.003$ | $0.003 \pm (5 \times 10^{-5})$ |
| DiffEnc-8-nt | fixed | $2.789 \pm 0.004$ | $(1.6 \times 10^{-5}) \pm 0.0$ | $2.779 \pm 0.004$ | $0.010 \pm (1 \times 10^{-5})$ |
| | trainable | $2.786 \pm 0.004$ | $0.0009 \pm (1 \times 10^{-5})$ | $2.782 \pm 0.004$ | $0.003 \pm (3 \times 10^{-5})$ |

a t-test on whether the mean loss over random seeds is lower for DiffEnc-32-4 than for VDMv-32. Some samples from DiffEnc-8-2, DiffEnc-32-4, and VDMv-32 are shown in Fig. 3. More samples from DiffEnc-32-4 and VDMv-32 in Appendix T. See Fig. 4 in the appendix for examples of encoded MNIST images. DiffEnc-32-4 and VDMv-32 have similar FID scores as shown in Table 8.

For all models with a trainable encoder and fixed noise schedule, we see that the diffusion loss is the same or better than the VDM baseline (see Tables 2, 3 and 4 to 7). We interpret this as the trainable encoder being able to preserve the most important signal as $t \to 1$. This is supported by the results we get from the non-trainable encoder, which only removes signal, where the diffusion loss is always worse than the baseline. We also see that, for a fixed noise schedule, the latent loss of the trainable encoder model is always better than the VDM. When using a fixed noise schedule, the minimal and maximal SNR is set to ensure small reconstruction and latent losses. It is therefore natural that the diffusion loss (the part dependent on how well the model can predict the noisy image), is the part that dominates the total loss. This means that a lower latent loss does not necessarily have a considerable impact on the total loss: For fixed noise schedule, the DiffEnc-8-2 models on MNIST and CIFAR-10 and the DiffEnc-32-8 model on ImageNet32 all have smaller latent loss than their VDMv counterparts, but since the diffusion loss is the same, the total loss does not show a significant change. However, Lin et al. (2023) pointed out that a high latent loss might lead to poor generated samples. Therefore, it might be relevant to train a model which has a lower latent loss than another model, if it can achieve the same diffusion loss. From Tables 3 and 4, we see that this is possible using a fixed noise schedule and a small trainable encoder. For results on a larger encoder with a small model see Appendix U.

We only saw an improvement in diffusion loss on the large models trained on CIFAR-10, and not on the small models. Since ImageNet32 is more complex than CIFAR-10, and we did not see an improvement in diffusion loss for the models on ImageNet32, a larger model might be needed on this dataset to see an improvement in diffusion loss. This would be interesting to test in future work.

For the trainable noise schedule, the mean total losses of the models are all lower than or equal to their fixed-schedule counterparts. Thus, all models can make some use use of this additional flexibility. For the fixed noise schedule, the reconstruction loss is the same for all three types of models, due to how our encoder is parameterized.

## 6 RELATED WORK

*DDPM:* Sohl-Dickstein et al. (2015) defined score-based diffusion models inspired by nonequilibrium thermodynamics. Ho et al. (2020) showed that diffusion models 1) are equivalent to score-based models and 2) can be viewed as hierarchical variational autoencoders with a diffusion encoder and parameter sharing in the generative hierarchy. Song et al. (2020b) defined diffusion models using an SDE.

*DDPM with encoder:* To the best of our knowledge, only few previous papers consider modifications of the diffusion process encoder. Implicit non-linear diffusion models (Kim et al., 2022b) use an invertible non-linear time-dependent map, $h$, to bring the data into a latent space where they do linear diffusion. $h$ can be compared to our encoder, however, we do not enforce the encoder to be invertible. Blurring diffusion models (Hoogeboom & Salimans, 2022; Rissanen et al., 2022) combines the

added noise with a blurring of the image dependent on the timestep. This blurring can be seen as a Gaussian encoder with a mean which is linear in the data, but with a not necessarily iid noise. The encoder parameters are set by the heat dissipation basis (the discrete cosine transform) and time. Our encoder is a learned non-linear function of the data and time and therefore more general than blurring. Daras et al. (2022) propose introducing a more general linear corruption process, where both blurring and masking for example can be added before the noise. Latent diffusion (Rombach et al., 2022) uses a learned depth-independent encoder/decoder to map deterministically between the data and a learned latent space and perform the diffusion in the latent space. Abstreiter et al. (2021) and Preechakul et al. (2022) use an additional encoder that computes a small semantic representation of the image. This representation is then used as conditioning in the diffusion model and is therefore orthogonal to our work. Singhal et al. (2023) propose to learn the noising process: for $\mathbf{z}_t = \alpha_t \mathbf{x} + \beta_t \epsilon$, they propose to learn $\alpha_t$ and $\beta_t$.

*Concurrent work:* Bartosh et al. (2023) also propose to add a time-dependent transformation to the data in the diffusion model. However, there is a difference in the target for the predictive function, since in our case $\hat{\mathbf{x}}_{\boldsymbol{\theta}}$ predicts the transformed data, $\mathbf{x}_{\phi}$, while in their case $\hat{\mathbf{x}}_{\boldsymbol{\theta}}$ predicts data $\mathbf{x}'$ such that the transformation of $\mathbf{x}'$, $f_{\phi}(\mathbf{x}', t)$, is equal to the transformation of the real data $f_{\phi}(\mathbf{x}, t)$. This might, according to their paper, make the prediction model learn something within the data distribution even for $t$ close to 1.

*Learned generative process variance:* Both Nichol & Dhariwal (2021) and Dhariwal & Nichol (2021) learn the generative process variance, $\sigma_P$. Dhariwal & Nichol (2021) observe that it allows for sampling with fewer steps without a large drop in sample quality and Nichol & Dhariwal (2021) argue that it could have a positive effect on the likelihood. Neither of these works are in a continuous-time setting, which is the setting we derived our theoretical results for.

## 7 LIMITATIONS AND FUTURE WORK

As shown above, adding a trained time-dependent encoder can improve the likelihood of a diffusion model, at the cost of a longer training time. Although our approach does not increase sampling time, it must be noted that sampling is still significantly slower than, e.g., for generative adversarial networks (Goodfellow et al., 2014). Techniques for more efficient sampling in diffusion models (Watson et al., 2021; Salimans & Ho, 2022; Song et al., 2020a; Lu et al., 2022; Berthelot et al., 2023; Luhman & Luhman, 2021; Liu et al., 2022) can be directly applied to our method.

Introducing the trainable encoder opens up an interesting new direction for representation learning. It should be possible to distill the time-dependent transformations to get smaller time-dependent representations of the images. It would be interesting to see what such representations could tell us about the data. It would also be interesting to explore whether adding conditioning to the encoder will lead to different transformations for different classes of images.

As shown in (Theis et al., 2015) likelihood and visual quality of samples are not directly linked. Thus it is important to choose the application of the model based on the metric it was trained to optimize. Since we show that our model can achieve good results when optimized to maximize likelihood, and likelihood is important in the context of semi-supervised learning, it would be interesting to use this kind of model for classification in a semi-supervised setting.

## 8 CONCLUSION

We presented DiffEnc, a generalization of diffusion models with a time-dependent encoder in the diffusion process. DiffEnc increases the flexibility of diffusion models while retaining the same computational requirements for sampling. Moreover, we theoretically derived the optimal variance of the generative process and proved that, in the continuous-time limit, it must be equal to the diffusion variance for the ELBO to be well-defined. We defer the investigation of its application to sampling or discrete-time training to future work. Empirically, we showed that DiffEnc can improve likelihood on CIFAR-10, and that the data transformation learned by the encoder is non-trivially dependent on the timestep. Interesting avenues for future research include applying improvements to diffusion models that are orthogonal to our proposed method, such as latent diffusion models, model distillation, classifier-free guidance, and different sampling strategies.

ETHICS STATEMENT

Since diffusion models have been shown to memorize training examples and since it is possible to extract these examples (Carlini et al., 2023), diffusion models pose a privacy and copyright risk especially if trained on data scraped from the internet. To the best of our knowledge our work neither improves nor worsens these security risks. Therefore, work still remains on how to responsibly deploy diffusion models with or without a time-dependent encoder.

REPRODUCIBILITY STATEMENT

The presented results are obtained using the setup described in Section 5. More details on models and training are discussed in Appendix Q. Code can be found on GitHub[1]. The Readme includes a description of setting up the environment with correct versioning. Scripts are supplied for recreating all results present in the paper. The main equations behind these results are Eqs. (19) and (20), which are the diffusion losses used when including our trainable and non-trainable encoder, respectively.

ACKNOWLEDGMENTS

This work was supported by the Danish Pioneer Centre for AI, DNRF grant number P1, and by the Ministry of Education, Youth and Sports of the Czech Republic through the e-INFRA CZ (ID:90254). OW's work was funded in part by the Novo Nordisk Foundation through the Center for Basic Machine Learning Research in Life Science (NNF20OC0062606). AC thanks the ELLIS PhD program for support.

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

# Appendix

## Table of Contents

## A  OVERVIEW OF DIFFUSION MODEL WITH AND WITHOUT ENCODER

The typical diffusion approach can be illustrated with the following diagram:

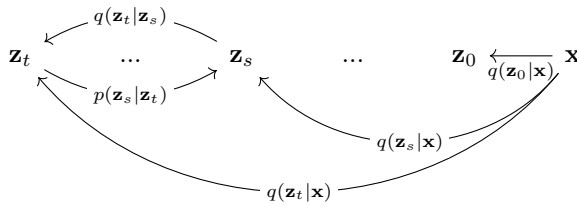

where $0 \leq s < t \leq 1$. We introduce an encoder $f_\phi : \mathcal{X} \times [0, 1] \rightarrow \mathcal{Y}$ with parameters $\phi$, that maps $\mathbf{x}$ and a time $t \in [0, 1]$ to a latent space $\mathcal{Y}$. In this work, $\mathcal{Y}$ has the same dimensions as the original image space. For brevity, we denote the encoded data as $\mathbf{x}_t \equiv f_\phi(\mathbf{x}, t)$. The following diagram illustrates the process including the encoder:

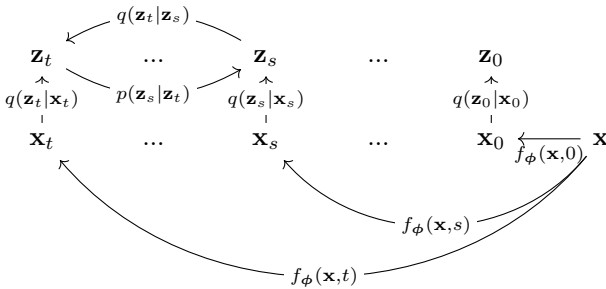

## B  PROOF THAT $\mathbf{z}_t$ GIVEN $\mathbf{x}$ HAS THE CORRECT FORM

Proof that we can write

$$q(\mathbf{z}_t|\mathbf{x}) = \mathcal{N}(\alpha_t \mathbf{x}_t, \sigma_t^2 \mathbf{I}) \tag{21}$$

for any $t$ when using the definition $q(\mathbf{z}_0|\mathbf{x}_0) = q(\mathbf{z}_0|\mathbf{x})$ and Eq. (11).

*Proof.* By induction:

The definition of $q(\mathbf{z}_0|\mathbf{x}_0) = q(\mathbf{z}_0|\mathbf{x})$ gives us our base case.

To take a step, we assume $q(\mathbf{z}_s|\mathbf{x}_s) = q(\mathbf{z}_s|\mathbf{x})$ can be written as

$$q(\mathbf{z}_s|\mathbf{x}) = \mathcal{N}(\alpha_s \mathbf{x}_s, \sigma_s^2 \mathbf{I}) \tag{22}$$

and take a $t > s$.

Then a sample from $q(\mathbf{z}_s|\mathbf{x})$ can be written as

$$\mathbf{z}_s = \alpha_s \mathbf{x}_s + \sigma_s \boldsymbol{\epsilon}_s \tag{23}$$

where $\boldsymbol{\epsilon}_s$ is from a standard normal distribution, $\mathcal{N}(\mathbf{0}, \mathbf{I})$ and a sample from $q(\mathbf{z}_t|\mathbf{z}_s, \mathbf{x}_t, \mathbf{x}_s) = q(\mathbf{z}_t|\mathbf{z}_s, \mathbf{x})$ can be written as

$$\mathbf{z}_t = \alpha_{t|s} \mathbf{z}_s + \alpha_t(\mathbf{x}_t - \mathbf{x}_s) + \sigma_{t|s} \boldsymbol{\epsilon}_{t|s} \tag{24}$$

where $\boldsymbol{\epsilon}_{t|s}$ is from a standard normal distribution, $\mathcal{N}(\mathbf{0}, \mathbf{I})$. Using the definition of $\mathbf{z}_s$, we get

$$\begin{aligned}
\mathbf{z}_t &= \alpha_{t|s}(\alpha_s \mathbf{x}_s + \sigma_s \boldsymbol{\epsilon}_s) + \alpha_t(\mathbf{x}_t - \mathbf{x}_s) + \sigma_{t|s} \boldsymbol{\epsilon}_{t|s} \\
&= \alpha_t \mathbf{x}_s + \alpha_{t|s} \sigma_s \boldsymbol{\epsilon}_s + \alpha_t \mathbf{x}_t - \alpha_t \mathbf{x}_s + \sigma_{t|s} \boldsymbol{\epsilon}_{t|s} \\
&= \alpha_t \mathbf{x}_t + \alpha_{t|s} \sigma_s \boldsymbol{\epsilon}_s + \sigma_{t|s} \boldsymbol{\epsilon}_{t|s}
\end{aligned} \tag{25}$$

Since $\alpha_{t|s}\sigma_s\boldsymbol{\epsilon}_s$ and $\sigma_{t|s}\boldsymbol{\epsilon}_{t|s}$ describe two normal distributions, a sample from the sum can be written as

$$\sqrt{\alpha_{t|s}^2\sigma_s^2 + \sigma_{t|s}^2}\boldsymbol{\epsilon}_t \tag{26}$$

where $\boldsymbol{\epsilon}_t$ is from a standard normal distribution, $\mathcal{N}(\mathbf{0}, \mathbf{I})$. So we can write our sample $\mathbf{z}_t$ as

$$
\begin{aligned}
\mathbf{z}_t &= \alpha_t\mathbf{x}_t + \sqrt{\alpha_{t|s}^2\sigma_s^2 + \sigma_{t|s}^2}\boldsymbol{\epsilon}_t \\
&= \alpha_t\mathbf{x}_t + \sqrt{\alpha_{t|s}^2\sigma_s^2 + \sigma_t^2 - \alpha_{t|s}^2\sigma_s^2}\boldsymbol{\epsilon}_t \\
&= \alpha_t\mathbf{x}_t + \sigma_t\boldsymbol{\epsilon}_t
\end{aligned}
\tag{27}
$$

Thus we get

$$q(\mathbf{z}_t|\mathbf{x}) = \mathcal{N}(\alpha_t\mathbf{x}_t, \sigma_t^2\mathbf{I}) \tag{28}$$

for any $0 \le t \le 1$. We have defined going from $\mathbf{x}$ to $\mathbf{z}_t$ as going through $f$. $\qquad\square$

## C    PROOF THAT THE REVERSE PROCESS HAS THE CORRECT FORM

To see that

$$q(\mathbf{z}_s|\mathbf{z}_t, \mathbf{x}_t, \mathbf{x}_s) = \mathcal{N}(\boldsymbol{\mu}_Q, \sigma_Q^2\mathbf{I}) \tag{29}$$

with

$$\sigma_Q^2 = \frac{\sigma_{t|s}^2\sigma_s^2}{\sigma_t^2} \tag{30}$$

and

$$\boldsymbol{\mu}_Q = \frac{\alpha_{t|s}\sigma_s^2}{\sigma_t^2}\mathbf{z}_t + \frac{\alpha_s\sigma_{t|s}^2}{\sigma_t^2}\mathbf{x}_t + \alpha_s(\mathbf{x}_s - \mathbf{x}_t) \tag{31}$$

is the right form for the reverse process, we take a sample $z_s$ from $q(\mathbf{z}_s|\mathbf{z}_t, \mathbf{x}_t, \mathbf{x}_s)$ and a sample $z_t$ from $q(z_t|x)$. These have the forms:

$$z_t = \alpha_t\mathbf{x}_t + \sigma_t\boldsymbol{\epsilon}_t \tag{32}$$

$$z_s = \frac{\alpha_{t|s}\sigma_s^2}{\sigma_t^2}\mathbf{z}_t + \frac{\alpha_s\sigma_{t|s}^2}{\sigma_t^2}\mathbf{x}_t + \alpha_s(\mathbf{x}_s - \mathbf{x}_t) + \sigma_Q^2\boldsymbol{\epsilon}_Q \tag{33}$$

We show that given $z_t$ we get $z_s$ from $q(z_s|x)$ as in Eq. (10).

$$z_s = \frac{\alpha_{t|s}\sigma_s^2}{\sigma_t^2}\left(\alpha_t\mathbf{x}_t + \sigma_t\boldsymbol{\epsilon}_t\right) + \frac{\alpha_s\sigma_{t|s}^2}{\sigma_t^2}\mathbf{x}_t + \alpha_s(\mathbf{x}_s - \mathbf{x}_t) + \sigma_Q^2\boldsymbol{\epsilon}_Q \tag{34}$$

$$= \frac{\alpha_t\alpha_{t|s}\sigma_s^2}{\sigma_t^2}\mathbf{x}_t + \frac{\alpha_s\left(\sigma_t^2 - \alpha_{t|s}^2\sigma_s^2\right)}{\sigma_t^2}\mathbf{x}_t + \alpha_s(\mathbf{x}_s - \mathbf{x}_t) + \frac{\alpha_{t|s}\sigma_s^2}{\sigma_t^2}\sigma_t\boldsymbol{\epsilon}_t + \sigma_Q^2\boldsymbol{\epsilon}_Q \tag{35}$$

$$\tag{36}$$

Since $\alpha_t = \alpha_s\alpha_{t|s}$ we have

$$z_s = \frac{\alpha_s\alpha_{t|s}^2\sigma_s^2}{\sigma_t^2}\mathbf{x}_t + \frac{\alpha_s\left(\sigma_t^2 - \alpha_{t|s}^2\sigma_s^2\right)}{\sigma_t^2}\mathbf{x}_t + \alpha_s(\mathbf{x}_s - \mathbf{x}_t) + \frac{\alpha_{t|s}\sigma_s^2}{\sigma_t^2}\sigma_t\boldsymbol{\epsilon}_t + \sigma_Q^2\boldsymbol{\epsilon}_Q \tag{37}$$

$$= \frac{\alpha_s\alpha_{t|s}^2\sigma_s^2}{\sigma_t^2}\mathbf{x}_t - \frac{\alpha_s\alpha_{t|s}^2\sigma_s^2}{\sigma_t^2}\mathbf{x}_t + \frac{\alpha_s\sigma_t^2}{\sigma_t^2}\mathbf{x}_t + \alpha_s(\mathbf{x}_s - \mathbf{x}_t) + \frac{\alpha_{t|s}\sigma_s^2}{\sigma_t^2}\sigma_t\boldsymbol{\epsilon}_t + \sigma_Q^2\boldsymbol{\epsilon}_Q \tag{38}$$

$$= \alpha_s\mathbf{x}_t + \alpha_s(\mathbf{x}_s - \mathbf{x}_t) + \frac{\alpha_{t|s}\sigma_s^2}{\sigma_t^2}\sigma_t\boldsymbol{\epsilon}_t + \sigma_Q^2\boldsymbol{\epsilon}_Q \tag{39}$$

$$= \alpha_s\mathbf{x}_s + \frac{\alpha_{t|s}\sigma_s^2}{\sigma_t^2}\sigma_t\boldsymbol{\epsilon}_t + \sigma_Q^2\boldsymbol{\epsilon}_Q \tag{40}$$

$$\tag{41}$$

We now use that $\sigma_Q^2 = \frac{\sigma_{t|s}^2 \sigma_s^2}{\sigma_t^2}$ and the sum rule of variances, $\sigma_{X+Y}^2 = \sigma_X^2 + \sigma_Y^2 + 2COV(X,Y)$, where the covariance is zero since $\epsilon_t$ and $\epsilon_Q$ are independent.

$$z_s = \alpha_s \mathbf{x}_s + \frac{\alpha_{t|s} \sigma_s^2}{\sigma_t^2} \sigma_t \epsilon_t + \sigma_Q^2 \epsilon_Q \tag{42}$$

$$= \alpha_s \mathbf{x}_s + \frac{\alpha_{t|s} \sigma_s^2}{\sigma_t} \epsilon_t + \frac{\sigma_{t|s} \sigma_s}{\sigma_t} \epsilon_Q \tag{43}$$

$$= \alpha_s \mathbf{x}_s + \sqrt{\frac{\alpha_{t|s}^2 \sigma_s^4}{\sigma_t^2} + \frac{\sigma_{t|s}^2 \sigma_s^2}{\sigma_t^2}} \epsilon_s \tag{44}$$

$$= \alpha_s \mathbf{x}_s + \sqrt{\frac{\alpha_{t|s}^2 \sigma_s^4}{\sigma_t^2} + \frac{(\sigma_t^2 - \alpha_{t|s}^2 \sigma_s^2) \sigma_s^2}{\sigma_t^2}} \epsilon_s \tag{45}$$

$$= \alpha_s \mathbf{x}_s + \sqrt{\frac{\sigma_t^2 \sigma_s^2}{\sigma_t^2}} \epsilon_s \tag{46}$$

$$= \alpha_s \mathbf{x}_s + \sigma_s \epsilon_s \tag{47}$$

Where $\epsilon_s$ is from a standard Gaussian distribution.

## D    THE LATENT AND RECONSTRUCTION LOSS

### D.1    LATENT LOSS

Since $q(z_1|\mathbf{x}) = \mathcal{N}(\alpha_1 \mathbf{x}_1, \sigma_1^2 \mathbf{I})$ and $p(z_1) = \mathcal{N}(\mathbf{0}, \mathbf{I})$, the latent loss, $D_{\mathrm{KL}}(q(z_1|\mathbf{x})||p(z_1))$, is the KL divergence of two normal distributions. For normal distributions $\mathcal{N}_0, \mathcal{N}_1$ with means $\boldsymbol{\mu}_0, \boldsymbol{\mu}_1$ and variances $\Sigma_0, \Sigma_1$, the KL divergence between them is given by

$$D_{\mathrm{KL}}(\mathcal{N}_0||\mathcal{N}_1) = \frac{1}{2} \left( \mathrm{tr}\left(\Sigma_1^{-1}\Sigma_0\right) - d + (\boldsymbol{\mu}_1 - \boldsymbol{\mu}_0)^T \Sigma_1^{-1} (\boldsymbol{\mu}_1 - \boldsymbol{\mu}_0) + \log\left(\frac{\det \Sigma_1}{\det \Sigma_0}\right) \right) , \tag{48}$$

where $d$ is the dimension. Therefore we have:

$$D_{\mathrm{KL}}(q(z_1|\mathbf{x})||p(z_1)) = \frac{1}{2} \left( \mathrm{tr}\left(\sigma_1^2 \mathbf{I}\right) - d + ||0 - \alpha_1 \mathbf{x}_1||^2 + \log\left(\frac{1}{\det \sigma_1^2 \mathbf{I}}\right) \right) \tag{49}$$

$$= \frac{1}{2} \left( ||\alpha_1 \mathbf{x}_1||^2 + d\left(\sigma_1^2 - \log \sigma_1^2 - 1\right) \right) \tag{50}$$

$$= \frac{1}{2} \left( \sum_{i=1}^{d} \left(\alpha_1^2 \mathbf{x}_{1,i}^2 + \sigma_1^2 - \log \sigma_1^2 - 1\right) \right) . \tag{51}$$

The last line is used in our implementation.

### D.2    RECONSTRUCTION LOSS

The reconstruction loss is given by

$$\mathcal{L}_0 = \mathbb{E}_{q(\mathbf{z}_0|\mathbf{x})} \left[ -\log p(\mathbf{x}|\mathbf{z}_0) \right] . \tag{52}$$

We make the simplifying assumption that $p(\mathbf{x}|\mathbf{z}_0)$ factorizes over the elements of $\mathbf{x}$. Let $x_i$ be the value of the $i$th dimension (i.e., pixel) of $\mathbf{x}$ and $z_{0,i}$ the corresponding pixel value of $\mathbf{z}_0$:

$$p(\mathbf{x}|\mathbf{z}_0) = \prod_i p(x_i|z_{0,i}) . \tag{53}$$

In our case of images, we assume the pixel values are independent given $\mathbf{z}_0$ and only dependent on the matching latent component. We construct $p(x_i|z_{0,i})$ from the variational distribution noting that

$$q(\mathbf{x}|\mathbf{z}_0) = \frac{q(\mathbf{z}_0|\mathbf{x})q(\mathbf{x})}{q(\mathbf{z}_0)} \tag{54}$$

and for high enough SNR at $t = 0$, $q(\mathbf{z}_0|\mathbf{x})$ will be very peaked around $\mathbf{z}_0 = \alpha_0 \mathbf{x}$. So we can choose

$$p(x_i|z_{0,i}) \propto q(z_{0,i}|x_i) = \mathcal{N}(z_{0,i}; \alpha_0 x_i, \sigma_0^2) \,, \tag{55}$$

where we normalize over all possible values of $x_i$. That is, let $v \in \{0, ..., 255\}$ be the possible pixel values of $x_i$, then for each $v$ we calculate the density $\mathcal{N}(\alpha_0 v, \sigma_0^2)$ at $z_{0,i}$ and then normalise over $v$ to get a categorical distribution $p(x_i|z_{0,i})$ that sums to 1.

## E  DIFFUSION LOSS

The diffusion loss is

$$\mathcal{L}_T(\mathbf{x}) = \sum_{i=1}^T \mathbb{E}_{q(z_{t(i)}|\mathbf{x})} D_{\mathrm{KL}}(q(\mathbf{z}_{s(i)}|\mathbf{z}_{t(i)}, \mathbf{x}) \| p_{\boldsymbol{\theta}}(\mathbf{z}_{s(i)}|\mathbf{z}_{t(i)})) \,, \tag{56}$$

where $s(i), t(i)$ are the values of $0 \le s < t \le 1$ corresponding to the $i$th timestep.

### E.1  ASSUMING NON-EQUAL VARIANCES IN THE DIFFUSION AND GENERATIVE PROCESSES

In this section we let

$$p_{\boldsymbol{\theta}}(\mathbf{z}_s|\mathbf{z}_t) = \mathcal{N}(\boldsymbol{\mu}_P, \sigma_P^2 \mathbf{I}) \tag{57}$$
$$q(\mathbf{z}_s|\mathbf{z}_t, \mathbf{x}) = \mathcal{N}(\boldsymbol{\mu}_Q, \sigma_Q^2 \mathbf{I}) \,, \tag{58}$$

where we might have $\sigma_P \ne \sigma_Q$. We then get for the KL divergence, where $d$ is the dimension,

$$D_{\mathrm{KL}}(q(\mathbf{z}_s|\mathbf{z}_t, \mathbf{x}) \| p_{\boldsymbol{\theta}}(\mathbf{z}_s|\mathbf{z}_t)) = D_{\mathrm{KL}}(\mathcal{N}(\mathbf{z}_s; \boldsymbol{\mu}_Q, \sigma_Q^2 \mathbf{I}) \| \mathcal{N}(\mathbf{z}_s; \boldsymbol{\mu}_P, \sigma_P^2 \mathbf{I}))$$

$$= \frac{1}{2} \left( \mathrm{tr}\left( \frac{\sigma_Q^2}{\sigma_P^2} \mathbf{I} \right) - d + (\boldsymbol{\mu}_P - \boldsymbol{\mu}_Q)^T \frac{1}{\sigma_P^2} \mathbf{I} (\boldsymbol{\mu}_P - \boldsymbol{\mu}_Q) + \log \frac{\det \sigma_P^2 \mathbf{I}}{\det \sigma_Q^2 \mathbf{I}} \right)$$

$$= \frac{1}{2} \left( d \frac{\sigma_Q^2}{\sigma_P^2} - d + \frac{1}{\sigma_P^2} \|\boldsymbol{\mu}_P - \boldsymbol{\mu}_Q\|_2^2 + \log \frac{(\sigma_P^2)^d}{(\sigma_Q^2)^d} \right)$$

$$= \frac{d}{2} \left( \frac{\sigma_Q^2}{\sigma_P^2} - 1 + \log \frac{\sigma_P^2}{\sigma_Q^2} \right) + \frac{1}{2\sigma_P^2} \|\boldsymbol{\mu}_P - \boldsymbol{\mu}_Q\|_2^2 \,. \tag{59}$$

If we define

$$w_t = \frac{\sigma_Q^2}{\sigma_P^2} \tag{60}$$

Then we can write the KL divergence as

$$D_{\mathrm{KL}}(q(\mathbf{z}_s|\mathbf{z}_t, \mathbf{x}) \| p_{\boldsymbol{\theta}}(\mathbf{z}_s|\mathbf{z}_t)) = \frac{d}{2} (w_t - 1 - \log w_t) + \frac{w_t}{2\sigma_Q^2} \|\boldsymbol{\mu}_P - \boldsymbol{\mu}_Q\|_2^2 \tag{61}$$

Using our definition of $\boldsymbol{\mu}_P$

$$\boldsymbol{\mu}_P = \frac{\alpha_{t|s}\sigma_s^2}{\sigma_t^2} \mathbf{z}_t + \frac{\alpha_s \sigma_{t|s}^2}{\sigma_t^2} \hat{\mathbf{x}}_{\boldsymbol{\theta}}(\lambda_t) + \alpha_s(\lambda_s - \lambda_t)(\sigma_t^2 \hat{\mathbf{x}}_{\boldsymbol{\theta}}(\lambda_t)) \tag{62}$$

we can rewrite the second term of the loss as:

$$\frac{w_t}{\sigma_Q^2} \|\boldsymbol{\mu}_P - \boldsymbol{\mu}_Q\|_2^2 \tag{63}$$

$$= \frac{w_t}{2\sigma_Q^2} \left\| \frac{\alpha_s \sigma_{t|s}^2}{\sigma_t^2} (\hat{\mathbf{x}}_{\boldsymbol{\theta}}(t) - \mathbf{x}_{\boldsymbol{\phi}}(\lambda_t)) + \alpha_s \left( (\lambda_s - \lambda_t)\sigma_t^2 \hat{\mathbf{x}}_{\boldsymbol{\theta}}(t) - (\mathbf{x}_{\boldsymbol{\phi}}(\lambda_s) - \mathbf{x}_{\boldsymbol{\phi}}(\lambda_t)) \right) \right\|_2^2 \,,$$

Where we have dropped the dependence on $\lambda$ from our notation of $\hat{\mathbf{x}}_{\boldsymbol{\theta}}(\lambda_t)$ and $\mathbf{x}_{\boldsymbol{\phi}}(\lambda_t)$, to make the equation fit on the page.

### E.2 ASSUMING EQUAL VARIANCES IN THE DIFFUSION AND GENERATIVE PROCESSES

If we let

$$p_{\boldsymbol{\theta}}(\mathbf{z}_s|\mathbf{z}_t) = \mathcal{N}(\boldsymbol{\mu}_P, \sigma_Q^2 \mathbf{I}) \tag{64}$$

$$q(\mathbf{z}_s|\mathbf{z}_t, \mathbf{x}) = \mathcal{N}(\boldsymbol{\mu}_Q, \sigma_Q^2 \mathbf{I}) \tag{65}$$

we get for the KL divergence

$$D_{\mathrm{KL}}(q(\mathbf{z}_s|\mathbf{z}_t, \mathbf{x}) \| p_{\boldsymbol{\theta}}(\mathbf{z}_s|\mathbf{z}_t)) = D_{\mathrm{KL}}(\mathcal{N}(\mathbf{z}_s; \boldsymbol{\mu}_Q, \sigma_Q^2 \mathbf{I}) \| \mathcal{N}(\mathbf{z}_s; \boldsymbol{\mu}_P, \sigma_Q^2 \mathbf{I}))$$

$$= \frac{1}{2\sigma_Q^2} \|\boldsymbol{\mu}_P - \boldsymbol{\mu}_Q\|_2^2$$

$$= \frac{1}{2\sigma_Q^2} \left\| \frac{\alpha_s \sigma_{t|s}^2}{\sigma_t^2} (\hat{\mathbf{x}}_{\boldsymbol{\theta}}(t) - \mathbf{x}_{\boldsymbol{\phi}}(\lambda_t)) + \alpha_s \left( (\lambda_s - \lambda_t) \sigma_t^2 \hat{\mathbf{x}}_{\boldsymbol{\theta}}(t) - (\mathbf{x}_{\boldsymbol{\phi}}(\lambda_s) - \mathbf{x}_{\boldsymbol{\phi}}(\lambda_t)) \right) \right\|_2^2. \tag{66}$$

## F  OPTIMAL VARIANCE FOR THE GENERATIVE MODEL

In this section, we compute the optimal variance $\sigma_P^2$ of the generative model in closed-form.

Consider the expectation over the data distribution of the KL divergence in the diffusion loss (Appendix E):

$$\mathbb{E}_{q(\mathbf{x}, \mathbf{z}_t)} \left[ D_{\mathrm{KL}}(q(\mathbf{z}_s|\mathbf{z}_t, \mathbf{x}) \| p_{\boldsymbol{\theta}}(\mathbf{z}_s|\mathbf{z}_t)) \right] = \frac{d}{2} \left( \frac{\sigma_Q^2}{\sigma_P^2} - 1 + \log \frac{\sigma_P^2}{\sigma_Q^2} \right) + \frac{1}{2\sigma_P^2} \mathbb{E}_{q(\mathbf{x}, \mathbf{z}_t)} \left[ \|\boldsymbol{\mu}_P - \boldsymbol{\mu}_Q\|_2^2 \right] \tag{67}$$

and differentiate it w.r.t. $\sigma_P^2$:

$$\frac{dD_{\mathrm{KL}}}{d\sigma_P^2} = \frac{d}{2} \left( -\frac{\sigma_Q^2}{\sigma_P^4} + \frac{1}{\sigma_Q^2} \frac{\sigma_Q^2}{\sigma_P^2} \right) - \frac{1}{2\sigma_P^4} \mathbb{E}_{q(\mathbf{x}, \mathbf{z}_t)} \left[ \|\boldsymbol{\mu}_P - \boldsymbol{\mu}_Q\|_2^2 \right] \tag{68}$$

$$= \frac{1}{2\sigma_P^4} \left( d\sigma_P^2 - d\sigma_Q^2 - \mathbb{E}_{q(\mathbf{x}, \mathbf{z}_t)} \left[ \|\boldsymbol{\mu}_P - \boldsymbol{\mu}_Q\|_2^2 \right] \right) \tag{69}$$

The derivative is zero when:

$$\sigma_P^2 = \sigma_Q^2 + \frac{1}{d} \mathbb{E}_{q(\mathbf{x}, \mathbf{z}_t)} \left[ \|\boldsymbol{\mu}_P - \boldsymbol{\mu}_Q\|_2^2 \right] \tag{70}$$

Since the second derivative of the KL at this value of $\sigma_P^2$ is positive, this is a minimum of the KL divergence.

## G  DIFFUSION LOSS IN CONTINUOUS TIME WITHOUT COUNTERTERM

In this section, we consider the DiffEnc diffusion process with mean shift term, coupled with the original VDM generative process (see Section 3). We show that in the continuous-time limit the optimal variance $\sigma_P^2$ tends to $\sigma_Q^2$ and the resulting diffusion loss simplifies to the standard VDM diffusion loss. We finally derive the diffusion loss in the continuous-time limit.

We start by rewriting the diffusion loss as expectation, using constant step size $\tau \equiv 1/T$ and denoting $t_i \equiv i/T$:

$$\mathcal{L}_T(\mathbf{x}) = T \, \mathbb{E}_{i \sim U\{1, T\}} \mathbb{E}_{q(\mathbf{z}_{t_i}|\mathbf{x})} \left[ D_{\mathrm{KL}}(q(\mathbf{z}_{t_i - \tau}|\mathbf{z}_{t_i}, \mathbf{x}) \| p_{\boldsymbol{\theta}}(\mathbf{z}_{t_i - \tau}|\mathbf{z}_{t_i})) \right] \tag{71}$$

$$= T \, \mathbb{E}_{t \sim U\{\tau, 2\tau, \dots, 1\}} \mathbb{E}_{q(\mathbf{z}_t|\mathbf{x})} \left[ D_{\mathrm{KL}}(q(\mathbf{z}_{t - \tau}|\mathbf{z}_t, \mathbf{x}) \| p_{\boldsymbol{\theta}}(\mathbf{z}_{t - \tau}|\mathbf{z}_t)) \right], \tag{72}$$

where we dropped indices and directly sample the discrete rv $t$.

The KL divergence can be calculated in closed form (Appendix E) because all distributions are Gaussian:

$$D_{\mathrm{KL}}(q(\mathbf{z}_{t - \tau}|\mathbf{z}_t, \mathbf{x}) \| p_{\boldsymbol{\theta}}(\mathbf{z}_{t - \tau}|\mathbf{z}_t)) = \frac{d}{2} (w_t - 1 - \log w_t) + \frac{w_t}{2\sigma_Q^2} \|\boldsymbol{\mu}_P - \boldsymbol{\mu}_Q\|_2^2, \tag{73}$$

where we have defined the weighting function

$$w_t = \frac{\sigma_{Q,t}^2}{\sigma_{P,t}^2} \tag{74}$$

Insert this in the diffusion loss:

$$\mathcal{L}_T(\mathbf{x}) = \mathbb{E}_{t \sim U\{\tau, 2\tau, \ldots, 1\}} \left[ \frac{d}{2\tau}(w_t - 1 - \log w_t) + \frac{1}{\tau} \mathbb{E}_{q(\mathbf{z}_t|\mathbf{x})} \left[ \frac{w_t}{2\sigma_Q^2} \|\boldsymbol{\mu}_P - \boldsymbol{\mu}_Q\|_2^2 \right] \right] \tag{75}$$

Given the optimal value for the noise variance in the generative model derived above (Appendix F):

$$\sigma_P^2 = \sigma_Q^2 + \frac{1}{d} \mathbb{E}_{q(\mathbf{x}, \mathbf{z}_t)} \left[ \|\boldsymbol{\mu}_P - \boldsymbol{\mu}_Q\|_2^2 \right]$$

we get the optimal $w_t$:

$$w_t^{-1} = 1 + \frac{1}{\sigma_Q^2 d} \mathbb{E}_{q(\mathbf{x}, \mathbf{z}_t)} \left[ \|\boldsymbol{\mu}_P - \boldsymbol{\mu}_Q\|_2^2 \right] \ .$$

Using the following definitions:

$$\boldsymbol{\mu}_Q = \frac{\alpha_{t|s}\sigma_s^2}{\sigma_t^2}\mathbf{z}_t + \frac{\alpha_s \sigma_{t|s}^2}{\sigma_t^2}\mathbf{x}_t + \alpha_s(\mathbf{x}_s - \mathbf{x}_t)$$

$$\boldsymbol{\mu}_P = \frac{\alpha_{t|s}\sigma_s^2}{\sigma_t^2}\mathbf{z}_t + \frac{\alpha_s \sigma_{t|s}^2}{\sigma_t^2}\hat{\mathbf{x}}_{\boldsymbol{\theta}}(\mathbf{z}_t, t)$$

$$\sigma_Q^2 = \frac{\sigma_{t|s}^2 \sigma_s^2}{\sigma_t^2}$$

$$\sigma_{t|s}^2 = \sigma_t^2 - \frac{\alpha_t^2}{\alpha_s^2}\sigma_s^2$$

and these intermediate results:

$$\frac{1}{\sigma_Q^2} \frac{\alpha_s^2 \sigma_{t|s}^4}{\sigma_t^4} = \text{SNR}(s) - \text{SNR}(t)$$

$$\frac{\sigma_{t|s}^2}{\sigma_t^2} = 1 - \frac{\alpha_t^2 \sigma_s^2}{\alpha_s^2 \sigma_t^2} = \frac{\text{SNR}(s) - \text{SNR}(t)}{\text{SNR}(s)}$$

we can write:

$$\frac{\|\boldsymbol{\mu}_P - \boldsymbol{\mu}_Q\|_2^2}{2\sigma_Q^2} = \frac{1}{2\sigma_Q^2} \left\| \frac{\alpha_s \sigma_{t|s}^2}{\sigma_t^2}\mathbf{x}_t + \alpha_s(\mathbf{x}_s - \mathbf{x}_t) - \frac{\alpha_s \sigma_{t|s}^2}{\sigma_t^2}\hat{\mathbf{x}}_{\boldsymbol{\theta}}(\mathbf{z}_t, t) \right\|_2^2$$

$$= \frac{1}{2\sigma_Q^2} \frac{\alpha_s^2 \sigma_{t|s}^4}{\sigma_t^4} \left\| \mathbf{x}_t + \frac{\sigma_t^2}{\sigma_{t|s}^2}(\mathbf{x}_s - \mathbf{x}_t) - \hat{\mathbf{x}}_{\boldsymbol{\theta}}(\mathbf{z}_t, t) \right\|_2^2$$

$$= -\frac{1}{2}\Delta\text{SNR} \left\| \mathbf{x}_t + \frac{\text{SNR}(s)}{\Delta\text{SNR}}\Delta\mathbf{x} - \hat{\mathbf{x}}_{\boldsymbol{\theta}}(\mathbf{z}_t, t) \right\|_2^2$$

where we used the shorthand $\Delta\text{SNR} \equiv \text{SNR}(t) - \text{SNR}(s)$ and $\Delta\mathbf{x} \equiv \mathbf{x}_t - \mathbf{x}_s$.

**The optimal $w_t$ tends to 1.** The optimal $w_t$ can be rewritten as follows:

$$w_t^{-1} = 1 + \frac{1}{\sigma_Q^2 d} \mathbb{E}_{q(\mathbf{x}, \mathbf{z}_t)} \left[ \|\boldsymbol{\mu}_P - \boldsymbol{\mu}_Q\|_2^2 \right] \tag{76}$$

$$= 1 - \frac{\Delta\text{SNR}}{d} \mathbb{E}_{q(\mathbf{x}, \mathbf{z}_t)} \left[ \left\| \mathbf{x}_t + \frac{\text{SNR}(s)}{\Delta\text{SNR}}\Delta\mathbf{x} - \hat{\mathbf{x}}_{\boldsymbol{\theta}}(\mathbf{z}_t, t) \right\|_2^2 \right] \tag{77}$$

As $T \to \infty$, or equivalently $s \to t$ and $\tau = s - t \to 0$, the optimal $w_t$ tends to 1, corresponding to the unweighted case (forward and backward variance are equal).

**The first term of diffusion loss tends to zero.** Define:

$$\nu = \frac{\Delta \text{SNR}}{d} \mathbb{E}_{q(\mathbf{x}, \mathbf{z}_t)} \left[ \left\| \mathbf{x}_t + \frac{\text{SNR}(s)}{\Delta \text{SNR}} \Delta \mathbf{x} - \hat{\mathbf{x}}_{\boldsymbol{\theta}}(\mathbf{z}_t, t) \right\|_2^2 \right]$$

such that the optimal $w_t$ is given by

$$w_t^{-1} = 1 - \nu$$

Then we are interested in the term

$$w_t - 1 - \log w_t = \frac{\nu}{1 - \nu} + \log(1 - \nu)$$

As $\tau \to 0$, we have

$$\Delta \text{SNR} = \tau \frac{d\text{SNR}(t)}{dt} + \mathcal{O}(\tau^2)$$

$$\Delta \mathbf{x} = \tau \frac{d\mathbf{x}_{\boldsymbol{\phi}}(\lambda_t)}{dt} + \mathcal{O}(\tau^2)$$

$$\nu = \frac{\tau}{d} \frac{d\text{SNR}(t)}{dt} \mathbb{E}_{q(\mathbf{x}, \mathbf{z}_t)} \left[ \left\| \mathbf{x}_t + \frac{d\mathbf{x}_{\boldsymbol{\phi}}(\lambda_t)}{d \log \text{SNR}} - \hat{\mathbf{x}}_{\boldsymbol{\theta}}(\mathbf{z}_t, t) \right\|_2^2 \right] + \mathcal{O}(\tau^2)$$

Since $\nu \to 0$, we can write a series expansion around $\nu = 0$:

$$w_t - 1 - \log w_t = \frac{1}{2}\nu^2 + \mathcal{O}(\nu^3)$$

$$= \frac{1}{2} \left( \frac{\tau}{d} \frac{d\text{SNR}(t)}{dt} \mathbb{E}_{q(\mathbf{x}, \mathbf{z}_t)} \left[ \left\| \mathbf{x}_t + \frac{d\mathbf{x}_{\boldsymbol{\phi}}(\lambda_t)}{d \log \text{SNR}} - \hat{\mathbf{x}}_{\boldsymbol{\theta}}(\mathbf{z}_t, t) \right\|_2^2 \right] \right)^2 + \mathcal{O}(\tau^3)$$

$$= \mathcal{O}(\tau^2)$$

*The first term of the weighted diffusion loss $\mathcal{L}_T$ is then 0*, since as $\tau \to 0$ we get:

$$\frac{d}{2\tau} \mathbb{E}_{t \sim U\{\tau, 2\tau, \ldots, 1\}} \left[ w_t - 1 - \log w_t \right] = \mathcal{O}(\tau) \tag{78}$$

Note that, had we simply used $w_t = 1 + \mathcal{O}(\tau)$, we would only be able to prove that this term in the loss is finite, but not whether it is zero. Here, we showed that the additional term in the loss actually tends to zero as $T \to \infty$.

In fact, we can also observe that, if $\sigma_P \neq \sigma_Q$ and therefore $w_t - 1 - \log w_t > 0$, the first term in the diffusion loss diverges in the continuous-time limit, so the ELBO is not well-defined.

**Continuous-time limit of the diffusion loss.** We saw that, as $\tau \to 0$, $w_t \to 1$ and the first term in the diffusion loss $\mathcal{L}_T$ tends to zero. The limit of $\mathcal{L}_T$ then becomes

$$\mathcal{L}_\infty(\mathbf{x}) = \lim_{T \to \infty} \mathcal{L}_T(\mathbf{x}) \tag{79}$$

$$= \lim_{T \to \infty} \mathbb{E}_{t \sim U\{\tau, 2\tau, \ldots, 1\}} \mathbb{E}_{q(\mathbf{z}_t | \mathbf{x})} \left[ \frac{1}{2\tau \sigma_Q^2} \|\boldsymbol{\mu}_P - \boldsymbol{\mu}_Q\|_2^2 \right] \tag{80}$$

$$= \lim_{T \to \infty} \mathbb{E}_{t \sim U\{\tau, 2\tau, \ldots, 1\}} \mathbb{E}_{q(\mathbf{z}_t | \mathbf{x})} \left[ -\frac{1}{2\tau} \frac{d\text{SNR}(t)}{dt} \tau \left\| \mathbf{x}_t + \frac{\text{SNR}(t) \frac{d\mathbf{x}_{\boldsymbol{\phi}}(\lambda_t)}{dt} \tau}{\frac{d\text{SNR}(t)}{dt} \tau} - \hat{\mathbf{x}}_{\boldsymbol{\theta}}(\mathbf{z}_t, t) \right\|_2^2 \right] \tag{81}$$

$$= -\frac{1}{2} \mathbb{E}_{t \sim U(0,1)} \mathbb{E}_{q(\mathbf{z}_t | \mathbf{x})} \left[ \frac{d\text{SNR}(t)}{dt} \left\| \mathbf{x}_t + \frac{d\mathbf{x}_{\boldsymbol{\phi}}(\lambda_t)}{d \log \text{SNR}} - \hat{\mathbf{x}}_{\boldsymbol{\theta}}(\mathbf{z}_t, t) \right\|_2^2 \right] \tag{82}$$

## H  DIFFENC AS AN SDE

A diffusion model may be seen as a discretization of an SDE. The same is true for the depth dependent encoder model. The forward process Eq. (11) can be written as

$$\mathbf{z}_t = \alpha_{t|s}\mathbf{z}_s + \alpha_t(\mathbf{x}_\phi(t) - \mathbf{x}_\phi(s)) + \sigma_{t|s}\boldsymbol{\epsilon} \ . \tag{83}$$

Let $0 < \Delta t < 1$ such that $t = s + \Delta t$. If we consider the first term after the equality sign we see that

$$\frac{\alpha_t}{\alpha_s} = \frac{\alpha_s + \alpha_t - \alpha_s}{\alpha_s} \tag{84}$$

$$= 1 + \frac{\alpha_t - \alpha_s}{\alpha_s \Delta t}\Delta t \tag{85}$$

So we get that

$$\mathbf{z}_t - \mathbf{z}_s = \frac{\alpha_t - \alpha_s}{\alpha_t \Delta t}\mathbf{z}_s\Delta t + \alpha_t(\mathbf{x}_\phi(t) - \mathbf{x}_\phi(s)) + \sigma_{t|s}\boldsymbol{\epsilon} \tag{86}$$

Considering the second term, we get

$$\alpha_t(\mathbf{x}_\phi(t) - \mathbf{x}_\phi(s)) = \alpha_t\frac{\mathbf{x}_\phi(t) - \mathbf{x}_\phi(s)}{\Delta t}\Delta t \tag{87}$$

So if we define

$$f_{\Delta t}(\mathbf{z}_s, s) = \frac{\alpha_t - \alpha_s}{\alpha_t \Delta t}\mathbf{z}_s + \alpha_t\frac{\mathbf{x}_\phi(t) - \mathbf{x}_\phi(s)}{\Delta t} \tag{88}$$

we can write

$$\mathbf{z}_t - \mathbf{z}_s = f_{\Delta t}(\mathbf{z}_s, s)\Delta t + \sigma_{t|t}\boldsymbol{\epsilon} \tag{89}$$

We will now consider $\sigma_{t|t}^2$ to be able to rewrite $\sigma_{t|s}\boldsymbol{\epsilon}$

$$\sigma_{t|t}^2 = \sigma_t^2 - \frac{\alpha_t^2}{\alpha_s^2}\sigma_s^2 \tag{90}$$

$$= \alpha_t^2\left(\frac{\sigma_t^2}{\alpha_t^2} - \frac{\sigma_s^2}{\alpha_s^2}\right) \tag{91}$$

$$= \alpha_t^2\left(\frac{\sigma_t^2}{\alpha_t^2} - \frac{\sigma_s^2}{\alpha_s^2}\right)\frac{\Delta t}{\Delta t} \tag{92}$$

Thus, if we define

$$g_{\Delta t}(s) = \sqrt{\alpha_t^2\left(\frac{\sigma_t^2}{\alpha_t^2} - \frac{\sigma_s^2}{\alpha_s^2}\right)\frac{1}{\Delta t}} \tag{93}$$

we can write

$$\mathbf{z}_t - \mathbf{z}_s = f_{\Delta t}(\mathbf{z}_s, s)\Delta t + g_{\Delta t}(s)\sqrt{\Delta t}\boldsymbol{\epsilon} \tag{94}$$

We can now take the limit $\Delta t \to 0$ using the definition $t = s + \Delta t$:

$$f_{\Delta t}(\mathbf{z}_s, s) \to \frac{1}{\alpha_s}\frac{d\alpha_s}{ds}\mathbf{z}_s + \alpha_s\frac{d\mathbf{x}_\phi(s)}{ds} = \frac{d\log\alpha_s}{ds}\mathbf{z}_s + \alpha_s\frac{d\mathbf{x}_\phi(s)}{ds} \tag{95}$$

$$g_{\Delta t}(s) \to \sqrt{\alpha_s^2\left(\frac{d\sigma_s^2/\alpha_s^2}{ds}\right)} \tag{96}$$

So if we use these limits to define the functions

$$\mathbf{f}(\mathbf{z}_t, t, \mathbf{x}) = \frac{1}{\alpha_t}\frac{d\alpha_t}{dt}\mathbf{z}_t + \alpha_t\frac{d\mathbf{x}_\phi(t)}{dt} \tag{97}$$

$$g(t) = \alpha_t\sqrt{\frac{d\sigma_t^2/\alpha_t^2}{dt}} \ . \tag{98}$$

we can write the forward stochastic process when using a time dependent encoder as

$$d\mathbf{z} = \mathbf{f}(\mathbf{z}_t, t, \mathbf{x})dt + g(t)d\mathbf{w} \tag{99}$$

where $d\mathbf{w}$ is the increment of a Wiener process over time $\Delta t$. The diffusion process of DiffEnc in the continuous-time limit is therefore similar to the usual SDE for diffusion models (Song et al., 2020b), with an additional contribution to the drift term.

Given the drift and the diffusion coefficient we can write the generative model as a reverse-time SDE (Song et al., 2020b):

$$dz = \left[\mathbf{f}(\mathbf{z}_t, t, \mathbf{x}) - g^2(t)\nabla_{\mathbf{z}_t} \log p(\mathbf{z}_t)\right] dt + g(t)d\bar{\mathbf{w}} \,, \tag{100}$$

where $d\bar{\mathbf{w}}$ is a reverse-time Wiener process.

## I   MOTIVATION FOR CHOICE OF PARAMETERIZATION FOR THE ENCODER

As mentioned in Section 4, we would like our encoding to be helpful for the reconstruction loss at $t = 0$ and for the latent loss at $t = 1$. Multiplying the data with $\alpha_t$ will give us these properties, since it will be close to the identity at $t = 0$ and send everything to 0 at $t = 1$. Instead of just multiplying with $\alpha_t$, we choose to use $\alpha_t^2$, since we still get the desirable properties for $t = 0$ and $t = 1$, but it makes some of the mathematical expressions nicer (for example the derivative). Thus, we arrive at the non-trainable parameterization

$$\mathbf{x}_{\mathrm{nt}}(\lambda_t) = \alpha_t^2 \mathbf{x} \tag{101}$$

At $t = 1$, we see that all the values of $\mathbf{x}_{\mathrm{nt}}(\lambda_t)$ are very close to 0, which should be easy to approximate from $z_1 = \alpha_1^3 \mathbf{x} + \sigma_1 \boldsymbol{\epsilon}$, since it will just be the mean of the values in $z_1$. Note that this parameterization gives us a lower latent loss, since the values of $\alpha_t^3 \mathbf{x}$ are closer to zero than the values of $\alpha_t \mathbf{x}$. However, this is not the same as just using a smaller minimum $\lambda_t$ in the original formulation, since in the original formulation the diffusion model would still be predicting $\mathbf{x}$ and not $\alpha_t^2 \mathbf{x} \approx 0$ at $t = 1$. There is still a problem with this formulation, since if we look at what happens between $t = 0$ and $t = 1$ we see that at some point, we will be attempting to approximate $v_t = \alpha_t \boldsymbol{\epsilon} - \sigma_t \mathbf{x}_{\mathrm{nt}}(x, \lambda_t)$ from a very noisy $\mathbf{z}_t$ while the values of $\mathbf{x}_{\mathrm{nt}}(x, \lambda_t)$ are still very small. In other words, since $\mathbf{z}_t$ is a noisy version of $\mathbf{x}_{\mathrm{nt}}(x, \lambda_t)$ and $\mathbf{x}_{\mathrm{nt}}(x, \lambda_t)$ has very small values there will not be much signal, but as we move away from $t = 1$, 0 will also become a worse and worse approximation.

This is why we introduce the trainable encoder

$$\mathbf{x}_\phi(\lambda_t) = \mathbf{x} - \sigma_t^2 \mathbf{x} + \sigma_t^2 \mathbf{y}_\phi(\mathbf{x}, \lambda_t) \tag{102}$$
$$= \alpha_t^2 \mathbf{x} + \sigma_t^2 \mathbf{y}_\phi(\mathbf{x}, \lambda_t) \tag{103}$$

Here we allow the inner encoder $\mathbf{y}_\phi(\mathbf{x}, \lambda_t)$ to add signal dependent on the image at the same pace as we are removing signal via the $-\sigma_t^2 \mathbf{x}$ term. This should give us a better diffusion loss between $t = 0$ and $t = 1$, but still has $\mathbf{x}_\phi(\lambda_t)$ very close to $\mathbf{x}$ at $t = 0$.

## J   CONTINUOUS-TIME LIMIT OF THE DIFFUSION LOSS WITH AN ENCODER

### J.1   REWRITING THE LOSS USING SNR

We can express the KL divergence in terms of the SNR:

$$\mathrm{SNR}(t) = \frac{\alpha_t^2}{\sigma_t^2} \,. \tag{104}$$

We pull $\frac{\alpha_s \sigma_{t|s}^2}{\sigma_t^2}$ outside, expand $\sigma_Q^2$, and use the definition of the SNR to get:

$$\frac{1}{2\sigma_Q^2} \frac{\alpha_s^2 \sigma_{t|s}^4}{\sigma_t^4} = \frac{1}{2} \left(\mathrm{SNR}(s) - \mathrm{SNR}(t)\right) \tag{105}$$

We also see that

$$\frac{\sigma_{t|s}^2}{\sigma_t^2} = \frac{\sigma_t^2 - \alpha_{t|s}^2 \sigma_s^2}{\sigma_t^2} = 1 - \frac{\alpha_t^2 \sigma_s^2}{\alpha_s^2 \sigma_t^2} = 1 - \frac{\mathrm{SNR}(t)}{\mathrm{SNR}(s)} = \frac{\mathrm{SNR}(s) - \mathrm{SNR}(t)}{\mathrm{SNR}(s)} \,. \tag{106}$$

Inserting this back into Eq. (66), we get:

$$\frac{1}{2\sigma_Q^2}\|\boldsymbol{\mu}_P - \boldsymbol{\mu}_Q\|_2^2 \tag{107}$$

$$= \frac{1}{2}\left(\text{SNR}(s) - \text{SNR}(t)\right)\cdot \tag{108}$$

$$\left\|\hat{\mathbf{x}}_{\boldsymbol{\theta}}(\lambda_t) - \mathbf{x}_{\boldsymbol{\phi}}(\lambda_t) + \frac{\text{SNR}(s)\left((\lambda_s - \lambda_t)\sigma_t^2\hat{\mathbf{x}}_{\boldsymbol{\theta}}(\lambda_t) - (\mathbf{x}_{\boldsymbol{\phi}}(\lambda_s) - \mathbf{x}_{\boldsymbol{\phi}}(\lambda_t))\right)}{\text{SNR}(s) - \text{SNR}(t)}\right\|_2^2$$

The KL divergence is then:

$$D_{\text{KL}}(q(\mathbf{z}_s|\mathbf{z}_t, \mathbf{x})\|p_{\boldsymbol{\theta}}(\mathbf{z}_s|\mathbf{z}_t)) \tag{109}$$

$$= \frac{1}{2}\left(\text{SNR}(s) - \text{SNR}(t)\right)\cdot$$

$$\left\|\hat{\mathbf{x}}_{\boldsymbol{\theta}}(\lambda_t) - \mathbf{x}_{\boldsymbol{\phi}}(\lambda_t) + \frac{\text{SNR}(s)\left((\lambda_s - \lambda_t)\sigma_t^2\hat{\mathbf{x}}_{\boldsymbol{\theta}}(\lambda_t) - (\mathbf{x}_{\boldsymbol{\phi}}(\lambda_s) - \mathbf{x}_{\boldsymbol{\phi}}(\lambda_t))\right)}{\text{SNR}(s) - \text{SNR}(t)}\right\|_2^2$$

with $s = \frac{i-1}{T}$ and $t = \frac{i}{T}$.

## J.2 TAKING THE LIMIT

If we rewrite everything in the loss from Eq. (109) to be with respect to $\lambda_t$, we get

$$\mathcal{L}_T(\mathbf{x}) = \frac{T}{2}\mathbb{E}_{\boldsymbol{\epsilon}, i\sim U\{1,T\}}\left[\left(e^{\lambda_s} - e^{\lambda_t}\right)\cdot \right. \tag{110}$$

$$\left.\left\|\hat{\mathbf{x}}_{\boldsymbol{\theta}}(\lambda_t) - \mathbf{x}_{\boldsymbol{\phi}}(\lambda_t) + \frac{e^{\lambda_s}\left((\lambda_s - \lambda_t)\sigma_t^2\hat{\mathbf{x}}_{\boldsymbol{\theta}}(\lambda_t) - (\mathbf{x}_{\boldsymbol{\phi}}(\lambda_s) - \mathbf{x}_{\boldsymbol{\phi}}(\lambda_t))\right)}{e^{\lambda_s} - e^{\lambda_t}}\right\|_2^2\right]$$

Where $s = (i-1)/T$ and $t = i/T$. We now want to take the continuous limit. Outside the norm we get the derivative with respect to $t$, inside the norm, we want the derivative w.r.t. $\lambda_t$. First we consider

$$\frac{e^{\lambda_s} - e^{\lambda_t}}{\frac{1}{T}} \tag{111}$$

For $T \to \infty$ and see that

$$\frac{e^{\lambda_s} - e^{\lambda_t}}{\frac{1}{T}} \to -\frac{de^{\lambda_t}}{dt} = -e^{\lambda_t}\cdot\lambda_t' \tag{112}$$

Where $\lambda_t'$ is the derivative of $\lambda_t$ w.r.t. $t$. Inside the norm we get for $s \to t$ that

$$e^{\lambda_s}\frac{\lambda_t - \lambda_s}{-(e^{\lambda_t} - e^{\lambda_s})}\frac{-(\mathbf{x}_{\boldsymbol{\phi}}(\lambda_t) - \mathbf{x}_{\boldsymbol{\phi}}(\lambda_s))}{\lambda_t - \lambda_s} \to e^{\lambda_t}\frac{-1}{\frac{de^{\lambda_t}}{d\lambda_t}}\frac{-d\mathbf{x}_{\boldsymbol{\phi}}(\lambda_t)}{d\lambda_t} \tag{113}$$

$$= \frac{e^{\lambda_t}}{e^{\lambda_t}}\frac{d\mathbf{x}_{\boldsymbol{\phi}}(\lambda_t)}{d\lambda_t} \tag{114}$$

$$= \frac{d\mathbf{x}_{\boldsymbol{\phi}}(\lambda_t)}{d\lambda_t} \tag{115}$$

and

$$\frac{e^{\lambda_s}}{e^{\lambda_s} - e^{\lambda_t}}(\lambda_s - \lambda_t)\sigma_t^2\hat{\mathbf{x}}_{\boldsymbol{\theta}}(\lambda_t) = e^{\lambda_s}\frac{-(\lambda_t - \lambda_s)}{-(e^{\lambda_t} - e^{\lambda_s})}\sigma_t^2\hat{\mathbf{x}}_{\boldsymbol{\theta}}(\lambda_t) \tag{116}$$

$$\to e^{\lambda_t}\frac{1}{e^{\lambda_t}}\sigma_t^2\hat{\mathbf{x}}_{\boldsymbol{\theta}}(\lambda_t) = \sigma_t^2\hat{\mathbf{x}}_{\boldsymbol{\theta}}(\lambda_t) \tag{117}$$

So we get the loss

$$\mathcal{L}_{\infty}(\mathbf{x}) = -\frac{1}{2}\mathbb{E}_{\boldsymbol{\epsilon}, t\sim U[0,1]}\left[\lambda_t' e^{\lambda_t}\left\|\hat{\mathbf{x}}_{\boldsymbol{\theta}}(\lambda_t) - \mathbf{x}_{\boldsymbol{\phi}}(\lambda_t) + \sigma_t^2\hat{\mathbf{x}}_{\boldsymbol{\theta}}(\lambda_t) - \frac{d\mathbf{x}_{\boldsymbol{\phi}}(\lambda_t)}{d\lambda_t}\right\|_2^2\right] \tag{118}$$

## K    USING THE V PARAMETERIZATION IN THE DIFFENC LOSS

In the following subsections we describe the **v**-prediction parameterization (Salimans & Ho, 2022) and derive the **v**-prediction loss for the proposed model, DiffEnc. We start by defining:

$$\mathbf{v}_t = \alpha_t \boldsymbol{\epsilon} - \sigma_t \mathbf{x}_\phi(\lambda_t) \tag{119}$$

$$\hat{\mathbf{v}}_{\boldsymbol{\theta}}(\lambda_t) = \alpha_t \hat{\boldsymbol{\epsilon}}_{\boldsymbol{\theta}} - \sigma_t \hat{\mathbf{x}}_{\boldsymbol{\theta}}(\lambda_t) \tag{120}$$

which give us (see Appendix K.1):

$$\mathbf{x}_\phi(\lambda_t) = \alpha_t \mathbf{z}_t - \sigma_t \mathbf{v}_t \tag{121}$$

$$\hat{\mathbf{x}}_{\boldsymbol{\theta}}(\lambda_t) = \alpha_t \mathbf{z}_t - \sigma_t \hat{\mathbf{v}}_{\boldsymbol{\theta}}(\lambda_t) \tag{122}$$

where we learn the **v**-prediction function $\hat{\mathbf{v}}_{\boldsymbol{\theta}}(\lambda_t) = \hat{\mathbf{v}}_{\boldsymbol{\theta}}(\mathbf{z}_{\lambda_t}, \lambda_t)$. In Appendix K.2 we show that, using this parameterization in Eq. (18), the loss becomes:

$$\mathcal{L}_\infty(\mathbf{x}) = -\frac{1}{2}\mathbb{E}_{\boldsymbol{\epsilon}, t \sim U[0,1]} \left[ \lambda_t' \alpha_t^2 \left\| \mathbf{v}_\phi(\lambda_t) - \hat{\mathbf{v}}_{\boldsymbol{\theta}}(\lambda_t) + \sigma_t \hat{\mathbf{x}}_{\boldsymbol{\theta}}(\lambda_t) - \frac{1}{\sigma_t}\frac{d\mathbf{x}_\phi(\lambda_t)}{d\lambda_t} \right\|_2^2 \right]. \tag{123}$$

As shown in Appendix K.3, the diffusion loss for the trainable encoder from Eq. (15) becomes:

$$\mathcal{L}_\infty(\mathbf{x}) = -\frac{1}{2}\mathbb{E}_{\boldsymbol{\epsilon}, t \sim U[0,1]} \left[ \lambda_t' \alpha_t^2 \left\| \mathbf{v}_t - \hat{\mathbf{v}}_{\boldsymbol{\theta}} + \sigma_t \left( \hat{\mathbf{x}}_{\boldsymbol{\theta}}(\lambda_t) - \mathbf{x}_\phi(\lambda_t) + \mathbf{y}_\phi(\lambda_t) - \frac{d\mathbf{y}_\phi(\lambda_t)}{d\lambda_t} \right) \right\|_2^2 \right] \tag{124}$$

and for the non-trainable encoder:

$$\mathcal{L}_\infty(\mathbf{x}) = -\frac{1}{2}\mathbb{E}_{\boldsymbol{\epsilon}, t \sim U[0,1]} \left[ \lambda_t' \alpha_t^2 \left\| \mathbf{v}_t - \hat{\mathbf{v}}_{\boldsymbol{\theta}} + \sigma_t \left( \hat{\mathbf{x}}_{\boldsymbol{\theta}}(\lambda_t) - \mathbf{x}_\phi(\lambda_t) \right) \right\|_2^2 \right]. \tag{125}$$

Eqs. (124) and (125) are the losses we use in our experiments.

### K.1    REWRITING THE V PARAMETERIZATION

In the v parameterization of the loss from (Salimans & Ho, 2022), $v_t$ is defined as

$$v_t = \alpha_t \boldsymbol{\epsilon} - \sigma_t \mathbf{x} \tag{126}$$

We use the generalization

$$v_t = \alpha_t \boldsymbol{\epsilon} - \sigma_t \mathbf{x}_\phi(x, \lambda_t) \tag{127}$$

Note that since

$$\mathbf{x}_\phi(x, \lambda_t) = (z_t - \sigma_t \boldsymbol{\epsilon})/\alpha_t \tag{128}$$

and

$$\alpha_t^2 + \sigma_t^2 = 1 \tag{129}$$

we get

$$\mathbf{x}_\phi(x, \lambda_t) = (z_t - \sigma_t \boldsymbol{\epsilon})/\alpha_t \tag{130}$$

$$= ((\alpha_t^2 + \sigma_t^2)z_t - \sigma_t(\alpha_t^2 + \sigma_t^2)\boldsymbol{\epsilon})/\alpha_t \tag{131}$$

$$= \left( \alpha_t + \frac{\sigma_t^2}{\alpha_t} \right) z_t - \left( \sigma_t \alpha_t + \frac{\sigma_t^3}{\alpha_t} \right) \boldsymbol{\epsilon} \tag{132}$$

$$= \alpha_t z_t + \frac{\sigma_t^2}{\alpha_t}z_t - \sigma_t \alpha_t \boldsymbol{\epsilon} - \frac{\sigma_t^3}{\alpha_t}\boldsymbol{\epsilon} \tag{133}$$

$$= \alpha_t z_t - \sigma_t \left( \alpha_t \boldsymbol{\epsilon} - \frac{\sigma_t}{\alpha_t}z_t + \frac{\sigma_t^2}{\alpha_t}\boldsymbol{\epsilon} \right) \tag{134}$$

$$= \alpha_t z_t - \sigma_t \left( \alpha_t \boldsymbol{\epsilon} - \frac{\sigma_t}{\alpha_t}(z_t - \sigma_t \boldsymbol{\epsilon}) \right) \tag{135}$$

$$= \alpha_t z_t - \sigma_t \left( \alpha_t \boldsymbol{\epsilon} - \sigma_t \mathbf{x}_\phi(x, \lambda_t) \right) \tag{136}$$

$$= \alpha_t z_t - \sigma_t v_t \tag{137}$$

$$\tag{138}$$

So

$$\mathbf{x}_{\boldsymbol{\phi}}(x, \lambda_t) = \alpha_t z_t - \sigma_t v_t \tag{139}$$

Therefore we define

$$\hat{\mathbf{v}}_{\boldsymbol{\theta}}(\lambda_t) = \alpha_t \hat{\boldsymbol{\epsilon}}_{\boldsymbol{\theta}} - \sigma_t \hat{\mathbf{x}}_{\boldsymbol{\theta}}(\lambda_t) \tag{140}$$

which in the same way gives us

$$\hat{\mathbf{x}}_{\boldsymbol{\theta}}(\mathbf{z}_{\lambda_t}, \lambda_t) = \alpha_t z_t - \sigma_t \hat{\mathbf{v}}_{\boldsymbol{\theta}} \tag{141}$$

where we learn $\hat{\mathbf{v}}_{\boldsymbol{\theta}}$.

## K.2 V PARAMETERIZATION IN CONTINUOUS DIFFUSION LOSS

For the **v** parameterization we have

$$\mathbf{v}_{\boldsymbol{\phi}}(\lambda_t) = \alpha_t \boldsymbol{\epsilon} - \sigma_t \mathbf{x}_{\boldsymbol{\phi}}(\lambda_t) \tag{142}$$

where $\boldsymbol{\epsilon}$ is from a standard normal distribution, $\mathcal{N}(\mathbf{0}, \mathbf{I})$, and

$$\mathbf{x}_{\boldsymbol{\phi}}(\lambda_t) = \alpha_t z_t - \sigma_t \mathbf{v}_{\boldsymbol{\phi}}(\lambda_t) \tag{143}$$

So we will set

$$\hat{\mathbf{x}}_{\boldsymbol{\theta}}(\lambda_t) = \alpha_t z_t - \sigma_t \hat{\mathbf{v}}_{\boldsymbol{\theta}}(\lambda_t) \tag{144}$$

and

$$\hat{\mathbf{v}}_{\boldsymbol{\theta}}(\lambda_t) = \alpha_t \hat{\boldsymbol{\epsilon}}_{\boldsymbol{\theta}} - \sigma_t \hat{\mathbf{x}}_{\boldsymbol{\theta}}(\lambda_t) \tag{145}$$

where we learn $\hat{\mathbf{v}}_{\boldsymbol{\theta}}(\lambda_t)$. If we rewrite the second term within the square brackets of Eq. (118) using the **v** parameterization, we get:

$$\lambda'(t)e^{\lambda_t} \left\| \hat{\mathbf{x}}_{\boldsymbol{\theta}}(\lambda_t) - \mathbf{x}_{\boldsymbol{\phi}}(\lambda_t) + \sigma_t^2 \hat{\mathbf{x}}_{\boldsymbol{\theta}}(\lambda_t) - \frac{d\mathbf{x}_{\boldsymbol{\phi}}(\lambda_t)}{d\lambda_t} \right\|_2^2 \tag{146}$$

$$= \lambda'(t)e^{\lambda_t} \left\| \sigma_t \mathbf{v}_{\boldsymbol{\phi}}(\lambda_t) - \sigma_t \hat{\mathbf{v}}_{\boldsymbol{\theta}}(\lambda_t) + \sigma_t^2 \hat{\mathbf{x}}_{\boldsymbol{\theta}}(\lambda_t) - \frac{d\mathbf{x}_{\boldsymbol{\phi}}(\lambda_t)}{d\lambda_t} \right\|_2^2 \tag{147}$$

$$= \lambda'(t)\alpha_t^2 \left\| \mathbf{v}_{\boldsymbol{\phi}}(\lambda_t) - \hat{\mathbf{v}}_{\boldsymbol{\theta}}(\lambda_t) + \sigma_t \hat{\mathbf{x}}_{\boldsymbol{\theta}}(\lambda_t) - \frac{1}{\sigma_t} \frac{d\mathbf{x}_{\boldsymbol{\phi}}(\lambda_t)}{d\lambda_t} \right\|_2^2 \tag{148}$$

So we get the loss

$$\mathcal{L}_{\infty}(\mathbf{x}) = -\frac{1}{2} \mathbb{E}_{\boldsymbol{\epsilon}, t \sim U[0,1]} \left[ \lambda'(t)\alpha_t^2 \left\| \mathbf{v}_{\boldsymbol{\phi}}(\lambda_t) - \hat{\mathbf{v}}_{\boldsymbol{\theta}}(\lambda_t) + \sigma_t \hat{\mathbf{x}}_{\boldsymbol{\theta}}(\lambda_t) - \frac{1}{\sigma_t} \frac{d\mathbf{x}_{\boldsymbol{\phi}}(\lambda_t)}{d\lambda_t} \right\|_2^2 \right] \tag{149}$$

## K.3 V PARAMETERIZATION OF CONTINUOUS DIFFUSION LOSS WITH ENCODER

We recall our two parameterizations of the encoder

$$\mathbf{x}_{\boldsymbol{\phi}}(\lambda_t) = \mathbf{x} - \sigma_t^2 \mathbf{x} + \sigma_t^2 \mathbf{y}_{\boldsymbol{\phi}}(\mathbf{x}, \lambda_t) \tag{150}$$

$$= \alpha_t^2 \mathbf{x} + \sigma_t^2 \mathbf{y}_{\boldsymbol{\phi}}(\mathbf{x}, \lambda_t) \tag{151}$$

and

$$\mathbf{x}_{\mathrm{nt}}(\lambda_t) = \alpha_t^2 \mathbf{x} \tag{152}$$

We see that

$$\frac{d\mathbf{x}_{\boldsymbol{\phi}}(\lambda_t)}{d\lambda_t} = \alpha_t^2 \sigma_t^2 \mathbf{x} + \sigma_t^2 \frac{d\mathbf{y}_{\boldsymbol{\phi}}(\lambda_t)}{d\lambda_t} - \alpha_t^2 \sigma_t^2 \mathbf{y}_{\boldsymbol{\phi}} \tag{153}$$

and

$$\frac{d\mathbf{x}_{\mathrm{nt}}(\lambda_t)}{d\lambda_t} = \alpha_t^2 \sigma_t^2 \mathbf{x} \tag{154}$$

as mentioned before. We first consider the loss for our trainable encoder. Focusing on the part of Eq. (123) inside the norm, and dropping the dependencies on $\lambda_t$ for brevity, we get

$$\mathbf{v}_\phi - \hat{\mathbf{v}}_\theta + \sigma_t \hat{\mathbf{x}}_\theta - \frac{1}{\sigma_t} \frac{d\mathbf{x}_\phi(\lambda_t)}{d\lambda_t} \tag{155}$$

$$= \mathbf{v}_\phi - \hat{\mathbf{v}}_\theta + \sigma_t \hat{\mathbf{x}}_\theta - \frac{1}{\sigma_t} \left( \alpha_t^2 \sigma_t^2 \mathbf{x} + \sigma_t^2 \frac{d\mathbf{y}_\phi(\lambda_t)}{d\lambda_t} - \alpha_t^2 \sigma_t^2 \mathbf{y}_\phi \right) \tag{156}$$

$$= \mathbf{v}_\phi - \hat{\mathbf{v}}_\theta + \sigma_t \hat{\mathbf{x}}_\theta - \alpha_t^2 \sigma_t \mathbf{x} - \sigma_t \frac{d\mathbf{y}_\phi(\lambda_t)}{d\lambda_t} + \alpha_t^2 \sigma_t \mathbf{y}_\phi \tag{157}$$

$$= \mathbf{v}_\phi - \hat{\mathbf{v}}_\theta + \sigma_t \left( \hat{\mathbf{x}}_\theta - \alpha_t^2 \mathbf{x} - \frac{d\mathbf{y}_\phi(\lambda_t)}{d\lambda_t} + \alpha_t^2 \mathbf{y}_\phi \right) \tag{158}$$

$$= \mathbf{v}_\phi - \hat{\mathbf{v}}_\theta + \sigma_t \left( \hat{\mathbf{x}}_\theta - \alpha_t^2 \mathbf{x} + (1 - \sigma_t^2)\mathbf{y}_\phi - \frac{d\mathbf{y}_\phi(\lambda_t)}{d\lambda_t} \right) \tag{159}$$

$$= \mathbf{v}_\phi - \hat{\mathbf{v}}_\theta + \sigma_t \left( \hat{\mathbf{x}}_\theta - \alpha_t^2 \mathbf{x} - \sigma_t^2 \mathbf{y}_\phi + \mathbf{y}_\phi - \frac{d\mathbf{y}_\phi(\lambda_t)}{d\lambda_t} \right) \tag{160}$$

$$= \mathbf{v}_\phi - \hat{\mathbf{v}}_\theta + \sigma_t \left( \hat{\mathbf{x}}_\theta - \mathbf{x}_\phi + \mathbf{y}_\phi - \frac{d\mathbf{y}_\phi(\lambda_t)}{d\lambda_t} \right) \tag{161}$$

So for the trainable encoder, we get the loss

$$\mathcal{L}_\infty(\mathbf{x}) = -\frac{1}{2} \mathbb{E}_{\epsilon, t \sim U[0,1]} \tag{162}$$

$$\left[ \lambda'(t)\alpha_t^2 \left\| \mathbf{v}_t - \hat{\mathbf{v}}_\theta + \sigma_t \left( \hat{\mathbf{x}}_\theta(\lambda_t) - \mathbf{x}_\phi(\lambda_t) + \mathbf{y}_\phi(\lambda_t) - \frac{d\mathbf{y}_\phi(\lambda_t)}{d\lambda_t} \right) \right\|_2^2 \right]$$

For the non-trainable encoder, if we again focus on the part of Eq. (123) inside the norm, and dropping the dependencies on $\lambda_t$ for brevity, we get

$$\mathbf{v}_\phi - \hat{\mathbf{v}}_\theta + \sigma_t \hat{\mathbf{x}}_\theta - \frac{1}{\sigma_t} \frac{d\mathbf{x}_{\mathrm{nt}}(\lambda_t)}{d\lambda_t} \tag{163}$$

$$= \mathbf{v}_\phi - \hat{\mathbf{v}}_\theta + \sigma_t \hat{\mathbf{x}}_\theta - \frac{1}{\sigma_t} \left( \alpha_t^2 \sigma_t^2 \mathbf{x} \right) \tag{164}$$

$$= \mathbf{v}_\phi - \hat{\mathbf{v}}_\theta + \sigma_t \hat{\mathbf{x}}_\theta - \sigma_t \left( \alpha_t^2 \mathbf{x} \right) \tag{165}$$

$$= \mathbf{v}_\phi - \hat{\mathbf{v}}_\theta + \sigma_t \left( \hat{\mathbf{x}}_\theta - \alpha_t^2 \mathbf{x} \right) \tag{166}$$

$$= \mathbf{v}_\phi - \hat{\mathbf{v}}_\theta + \sigma_t \left( \hat{\mathbf{x}}_\theta - \mathbf{x}_{\mathrm{nt}} \right) \tag{167}$$

So for the non-trainable encoder, we get the loss

$$\mathcal{L}_\infty(\mathbf{x}) = -\frac{1}{2} \mathbb{E}_{\epsilon, t \sim U[0,1]} \left[ \lambda'(t)\alpha_t^2 \left\| \mathbf{v}_t - \hat{\mathbf{v}}_\theta + \sigma_t \left( \hat{\mathbf{x}}_\theta(\lambda_t) - \mathbf{x}_{\mathrm{nt}}(\lambda_t) \right) \right\|_2^2 \right] \tag{168}$$

## L  CONSIDERING LOSS FOR EARLY AND LATE TIMESTEPS

Let us consider what happens to the expression inside the norm from our loss Eq. (19) for $t$ close to zero. We see that since $\alpha_t \to 1$ and $\sigma_t \to 0$ for $t \to 0$ and $\hat{\mathbf{v}}_\theta = \alpha_t \hat{\epsilon}_\theta - \sigma_t \hat{\mathbf{x}}_\theta(\lambda_t)$, we get for the trainable encoder

$$\left\| \mathbf{v}_t - \hat{\mathbf{v}}_\theta + \sigma_t \left( \hat{\mathbf{x}}_\theta(\lambda_t) - \mathbf{x}_\phi(\lambda_t) + \mathbf{y}_\phi(\lambda_t) - \frac{d\mathbf{y}_\phi(\lambda_t)}{d\lambda_t} \right) \right\|_2^2 \tag{169}$$

$$\to \left\| \mathbf{v}_t - \hat{\mathbf{v}}_\theta \right\|_2^2 = \left\| \epsilon - \hat{\epsilon}_\theta \right\|_2^2 \tag{170}$$

and for the non-trainable encoder

$$\left\| \mathbf{v}_t - \hat{\mathbf{v}}_\theta + \sigma_t \left( \hat{\mathbf{x}}_\theta(\lambda_t) - \mathbf{x}_\phi(\lambda_t) \right) \right\|_2^2 \tag{171}$$

$$\to \left\| \mathbf{v}_t - \hat{\mathbf{v}}_\theta \right\|_2^2 = \left\| \epsilon - \hat{\epsilon}_\theta \right\|_2^2 \tag{172}$$

Table 4: Comparison of the different components of the loss for DiffEnc-8-2, DiffEnc-8-nt and VDMv-8 on MNIST. All quantities are in bits per dimension (BPD), with standard error, 5 seeds, 2M steps. Noise schedules are either fixed or with trainable endpoints.

| Model | Noise | Total | Latent | Diffusion | Reconstruction |
|-------|-------|-------|--------|-----------|----------------|
| VDMv-8 | fixed | $0.370 \pm 0.002$ | $0.0045 \pm 0.0$ | $0.360 \pm 0.002$ | $0.006 \pm (3 \times 10^{-5})$ |
| | trainable | $0.366 \pm 0.001$ | $0.0042 \pm (5 \times 10^{-5})$ | $0.361 \pm 0.003$ | $0.001 \pm (2 \times 10^{-5})$ |
| DiffEnc-8-2 | fixed | $0.367 \pm 0.001$ | $0.0009 \pm (3 \times 10^{-6})$ | $0.360 \pm 0.001$ | $0.006 \pm (3 \times 10^{-5})$ |
| | trainable | $0.363 \pm 0.002$ | $0.0064 \pm (8 \times 10^{-5})$ | $0.355 \pm 0.002$ | $0.001 \pm (2 \times 10^{-5})$ |
| DiffEnc-8-nt | fixed | $0.378 \pm 0.002$ | $\underline{1.6 \times 10^{-5}} \pm 0.0$ | $0.371 \pm 0.002$ | $0.006 \pm (3 \times 10^{-5})$ |
| | trainable | $0.373 \pm 0.001$ | $0.0021 \pm (3 \times 10^{-5})$ | $0.369 \pm 0.001$ | $0.002 \pm (5 \times 10^{-5})$ |

So we get the same objective as for the epsilon parameterization used in (Kingma et al., 2021) in both cases. On the other hand, since $\sigma_t \to 1$ as $t \to 1$, we get for the trainable encoder:

$$\left\| \mathbf{v}_t - \hat{\mathbf{v}}_{\boldsymbol{\theta}} + \sigma_t \left( \hat{\mathbf{x}}_{\boldsymbol{\theta}}(\lambda_t) - \mathbf{x}_{\boldsymbol{\phi}}(\lambda_t) + \mathbf{y}_{\boldsymbol{\phi}}(\lambda_t) - \frac{d\mathbf{y}_{\boldsymbol{\phi}}(\lambda_t)}{d\lambda_t} \right) \right\|_2^2 \tag{173}$$

$$\to \left\| \hat{\mathbf{x}}_{\boldsymbol{\theta}}(\lambda_t) - 2\mathbf{x}_{\boldsymbol{\phi}}(\lambda_t) + \mathbf{y}_{\boldsymbol{\phi}}(\lambda_t) - \frac{d\mathbf{y}_{\boldsymbol{\phi}}(\lambda_t)}{d\lambda_t} - \hat{\mathbf{v}}_{\boldsymbol{\theta}} \right\|_2^2 \tag{174}$$

Assuming $\mathbf{x}_{\boldsymbol{\phi}}(\lambda_t) \approx \hat{\mathbf{x}}_{\boldsymbol{\theta}}(\lambda_t)$, this loss is small at $t \approx 1$ if:

$$\hat{\mathbf{v}}_{\boldsymbol{\theta}} \approx -\mathbf{x}_{\boldsymbol{\phi}}(\lambda_t) + \mathbf{y}_{\boldsymbol{\phi}}(\lambda_t) - \frac{d\mathbf{y}_{\boldsymbol{\phi}}(\lambda_t)}{d\lambda_t} \tag{175}$$

$$= -\mathbf{x} + \sigma_t^2 \mathbf{x} - \sigma_t^2 \mathbf{y}_{\boldsymbol{\phi}}(\mathbf{x}, \lambda_t) + \mathbf{y}_{\boldsymbol{\phi}}(\lambda_t) - \frac{d\mathbf{y}_{\boldsymbol{\phi}}(\lambda_t)}{d\lambda_t} \tag{176}$$

$$\approx -\mathbf{x} + \mathbf{x} - \mathbf{y}_{\boldsymbol{\phi}}(\lambda_t) + \mathbf{y}_{\boldsymbol{\phi}}(\lambda_t) - \frac{d\mathbf{y}_{\boldsymbol{\phi}}(\lambda_t)}{d\lambda_t} \tag{177}$$

$$= -\frac{d\mathbf{y}_{\boldsymbol{\phi}}(\lambda_t)}{d\lambda_t} \tag{178}$$

So we are saying that at $t = 1$, $\hat{\mathbf{v}}_{\boldsymbol{\theta}} \approx -\frac{d\mathbf{y}_{\boldsymbol{\phi}}(\lambda_t)}{d\lambda_t}$. Thus the encoder should be able to guide the diffusion model. For the non-trainable encoder, we get

$$\left\| \mathbf{v}_t - \hat{\mathbf{v}}_{\boldsymbol{\theta}} + \sigma_t \left( \hat{\mathbf{x}}_{\boldsymbol{\theta}}(\lambda_t) - \mathbf{x}_{\boldsymbol{\phi}}(\lambda_t) \right) \right\|_2^2 \tag{179}$$

$$\to \left\| -\mathbf{x}_{\boldsymbol{\phi}}(\lambda_t) + \hat{\mathbf{x}}_{\boldsymbol{\theta}}(\lambda_t) + \hat{\mathbf{x}}_{\boldsymbol{\theta}}(\lambda_t) - \mathbf{x}_{\boldsymbol{\phi}}(\lambda_t) \right\|_2^2 \tag{180}$$

$$= \left\| 2\hat{\mathbf{x}}_{\boldsymbol{\theta}}(\lambda_t) - 2\mathbf{x}_{\boldsymbol{\phi}}(\lambda_t) \right\|_2^2 \tag{181}$$

So in this case, we are just saying that $\hat{\mathbf{x}}_{\boldsymbol{\theta}}(\lambda_t)$ should be close to $\mathbf{x}_{\boldsymbol{\phi}}(\lambda_t)$. However, note that since $\mathbf{x}_{\boldsymbol{\phi}}(\lambda_t) = \alpha_t^2 \mathbf{x}$, we have that $\hat{\mathbf{x}}_{\boldsymbol{\theta}}(\lambda_t) \approx \mathbf{x}_{\boldsymbol{\phi}}(\lambda_t) \approx 0$ for $t = 1$. So this is only saying that it should be easy to guess $\mathbf{x}_{\boldsymbol{\phi}}(\lambda_t) \approx 0$ for $t \approx 1$, but it will not help the diffusion model guessing the signal, since there is no signal left in this case.

## M  DETAILED LOSS COMPARISON FOR DIFFENC AND VDMV ON MNIST

Table 4 shows the average losses of the models trained on MNIST. We see the same pattern as for the small models trained on CIFAR-10: All models with a trainable encoder achieve the same or better diffusion loss than the VDMv model. For the fixed noise schedules the latent loss is always better for the DiffEnc models than for the VDMv, however for the trainable noise schedule, it seems the DiffEnc with a learned encoder sacrifices some latent loss to gain a better diffusion loss.

## N  DETAILED LOSS COMPARISON FOR DIFFENC-32-2 AND VDMV-32 ON CIFAR-10

To explore the significance of the encoder size, we trained a DiffEnc-32-2, that is, a large diffusion model with a smaller encoder, see Table 5. We see that after 2M steps the diffusion loss for the

Table 5: Comparison of the different components of the loss for DiffEnc-32-2 and VDMv-32 with fixed noise schedule on CIFAR-10. All quantities are in bits per dimension (BPD) with standard error over 3 seeds, comparison at 2M steps.

| Model | Total | Latent | Diffusion | Reconstruction |
|---|---|---|---|---|
| VDMv-32 | $2.666 \pm 0.002$ | $0.0012 \pm 0.0$ | $2.654 \pm 0.003$ | $\underline{0.01} \pm (4 \times 10^{-6})$ |
| DiffEnc-32-2 | $2.660 \pm 0.006$ | $\underline{0.0007} \pm (3 \times 10^{-6})$ | $2.649 \pm 0.006$ | $\underline{0.01} \pm (2 \times 10^{-6})$ |

Table 6: Comparison of the different components of the loss for DiffEnc-32-8 and VDMv-32 with fixed noise schedule on ImageNet32. All quantities are in bits per dimension (BPD) with standard error over 3 seeds, and models are trained for 1.5M steps.

| Model | Total | Latent | Diffusion | Reconstruction |
|---|---|---|---|---|
| VDMv-32 | $3.461 \pm 0.002$ | $0.0014 \pm 0.0$ | $3.449 \pm 0.002$ | $\underline{0.01} \pm (1 \times 10^{-5})$ |
| DiffEnc-32-8 | $3.461 \pm 0.002$ | $\underline{0.0007} \pm (9 \times 10^{-7})$ | $3.450 \pm 0.002$ | $\underline{0.01} \pm (1 \times 10^{-5})$ |

DiffEnc model is smaller than for the VDMv, however, not significantly so. When inspecting a plot of the losses of the models, the losses seem to be diverging, but one would have to train the DiffEnc-32-2 model for longer to be certain. We did not continue this experiment because of the large compute cost.

## O    DETAILED LOSS COMPARISON FOR DIFFENC AND VDMV ON IMAGENET32

On imagenet32, we see the same pattern in our experiments as for the small models on CIFAR-10 and MNIST, see Table 6. The diffusion loss is the same for the two models, but the latent loss is better for DiffEnc. Since ImageNet is more complex than CIFAR-10, we might need an even larger base diffusion model to achieve a difference in diffusion loss.

## P    FURTHER FUTURE WORK

Our approach could be combined with various existing methods, e.g., latent diffusion (Vahdat et al., 2021; Rombach et al., 2022) or discriminator guidance (Kim et al., 2022a). If one were to succeed in making the smaller representations from the encoder, one might also combine it with consistency regularization (Sinha & Dieng, 2021) to improve the learned representations.

## Q    MODEL STRUCTURE AND TRAINING

Code can be found on GitHub[2].

All our diffusion models use the same overall structure with $n$ ResNet blocks, then a middle block of 1 ResNet, 1 self attention and 1 ResNet block, and in the end $n$ more ResNet blocks. We train diffusion models with $n = 8$ on MNIST and CIFAR-10 and models with $n = 32$ on CIFAR-10 and ImageNet32. All ResNet blocks in the diffusion models preserve the dimensions of the original images (28x28 for MNIST, 32x32 for CIFAR-10, 32x32 for ImageNet32) and have 128 out channels for models on MNIST and CIFAR-10 and 256 out channels for models on ImageNet32 following (Kingma et al., 2021). We use both a fixed noise schedule with $\lambda_{max} = 13.3$ and $\lambda_{min} = -5$ and a trainable noise schedule where we learn $\lambda_{max}$ and $\lambda_{min}$.

For our encoder, we use a very similar overall structure as for the diffusion model. Here we have $m$ ResNet blocks, then a middle block of 1 ResNet, 1 self attention and 1 ResNet block, and in the end

---

[2]https://github.com/bemigini/DiffEnc

Table 7: Comparison of the different components of the loss for DiffEnc-8-4 and VDMv-8 on CIFAR-10 with fixed noise schedule after 1.3M steps. All quantities are in bits per dimension (BPD), with standard error, 3 seeds for DiffEnc-8-4, 5 seeds for VDMv-8.

| Model | Total | Latent | Diffusion | Reconstruction |
|---|---|---|---|---|
| VDMv-8 | $2.794 \pm 0.004$ | $0.0012 \pm 0.0$ | $2.782 \pm 0.004$ | $0.010 \pm (1 \times 10^{-5})$ |
| DiffEnc-8-4 | $2.789 \pm 0.002$ | $\underline{0.0006} \pm (2 \times 10^{-6})$ | $2.778 \pm 0.002$ | $0.010 \pm (1 \times 10^{-5})$ |

$m$ more ResNet blocks. However, for the encoder with $m = 2$, we use maxpooling after each of the first $m$ ResNet blocks and transposed convolution after the last $m$ ResNet blocks, for encoders with $m = 4$, we use maxpooling after every other of the first $m$ ResNet blocks and transposed convolution after every other of the last $m$ ResNet blocks and for encoders with $m = 8$, we use maxpooling after every fourth of the first $m$ ResNet blocks and transposed convolution after every fourth of the last $m$ ResNet blocks. Thus, for the encoder we downscale to and upscale from resolutions 14x14 and 7x7 on MNIST and 16x16 and 8x8 on CIFAR-10 and ImageNet32.

We do experiments with $n = 8$, $m = 2$ on MNIST and CIFAR-10, $n = 8$, $m = 2$ and $n = 32$, $m = 4$ on CIFAR-10 and $n = 32$, $m = 8$ on ImageNet32.

We trained 5 seeds for the small models ($n = 8$), except for the diffusion model size 8 encoder size 4 on CIFAR-10 where we trained 3 seeds. We trained 3 seeds for the large models ($n = 32$).

For models on MNIST and CIFAR-10 we used a batch size of 128 and no gradient clipping. For models on ImageNet32 we used a batch size of 256 and no gradient clipping.

## R  DATASETS

We considered three datasets:

- MNIST: The MNIST dataset (LeCun et al., 1998) as fetched by the tensorflow_datasets package[3]. 60,000 images were used for training and 10,000 images for test. License: Unknown.

- CIFAR-10: The CIFAR-10 dataset as fetched from the tensorflow_datasets package[4]. Originally collected by Krizhevsky et al. (2009). 50,000 images were used for training and 10,000 images for test. License: Unknown.

- ImageNet $32 \times 32$: The official downsampled version of ImageNet (Chrabaszcz et al., 2017) from the ImageNet website: `https://image-net.org/download-images.php`.

## S  ENCODER EXAMPLES ON MNIST

Fig. 4 provides an example of the encodings we get from MNIST when using DiffEnc with a learned encoder.

## T  SAMPLES FROM MODELS

Examples of samples from our large trained models, DiffEnc-32-4 and VDMv-32, can be seen in Fig. 5.

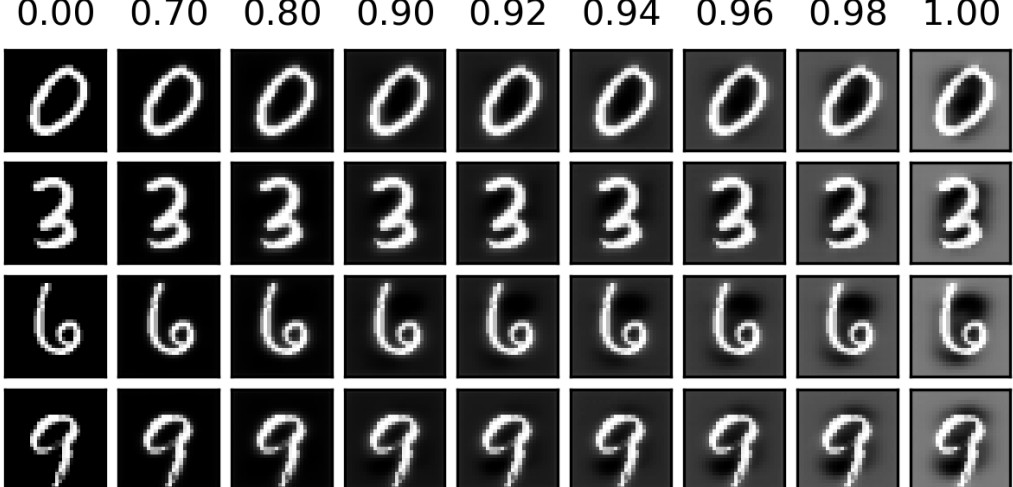

Figure 4: Encoded MNIST images from DiffEnc-8-2. Encoded images are close to the identity up to $t = 0.7$. From $t = 0.8$ to $t = 0.9$ the encoder slightly blurs the numbers, and from $t = 0.9$ it makes the background lighter, but keeps the high contrast in the middle of the image. Intuitively, the encoder improves the latent loss by bringing the average pixel value close to 0.

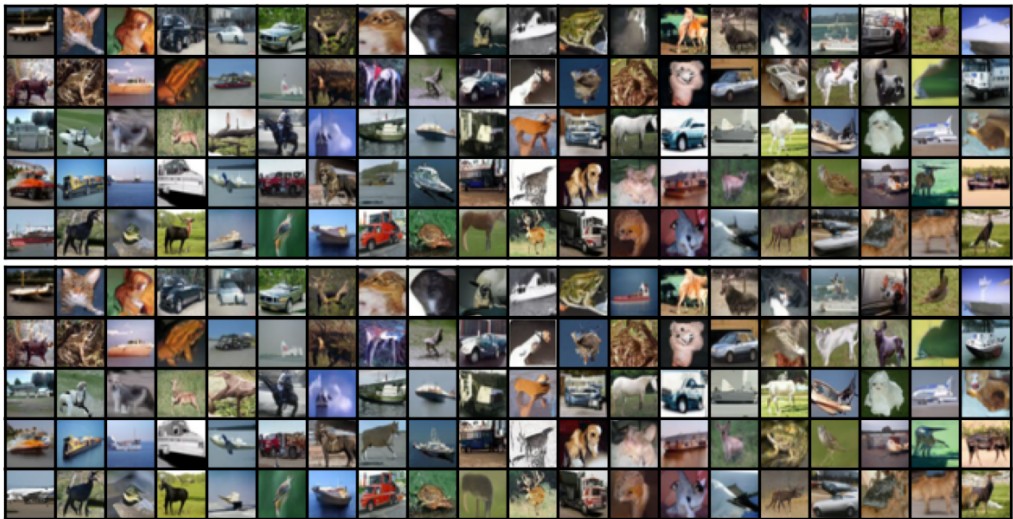

Figure 5: 100 unconditional samples from a DiffEnc-32-4 (above) and VDMv-32 (below) after 8 million training steps.

## U USING A LARGER ENCODER FOR A SMALL DIFFUSION MODEL

As we can observe in Table 7, when the encoder's size is increased, the average diffusion loss is slightly smaller than that of VDM, albeit not significantly. We propose the following two potential explanations for this phenomenon: (1) Longer training may be needed to achieve a significant difference. For DiffEnc-32-4 and VDMv-32, we saw different trends in the loss after about 2 million steps, where the loss of the DiffEnc models decreased more per step. However, it took more training with this trend to achieve a substantial divergence in diffusion loss. (2) a larger diffusion model may be required to fully exploit the encoder.

---

[3]https://www.tensorflow.org/datasets/catalog/cifar10
[4]https://www.tensorflow.org/datasets/catalog/cifar10

Table 8: Comparison of the mean FID scores with standard error for DiffEnc-32-4 and VDMv-32 on CIFAR-10 with fixed noise schedule after 8M steps. 3 seeds. We provide both FID scores on 10K and 50K samples and with respect to both train and test set.

| Model | FID 10K train | FID 10K test | FID 50K train | FID 50K test |
|-------|---------------|--------------|---------------|--------------|
| VDMv-32 | $14.8 \pm 0.2$ | $18.9 \pm 0.2$ | $11.2 \pm 0.2$ | $14.9 \pm 0.2$ |
| DiffEnc-32-4 | $14.6 \pm 0.8$ | $18.5 \pm 0.7$ | $11.1 \pm 0.8$ | $15.0 \pm 0.7$ |

## V    FID SCORES

Although we did not optimize our model for the visual quality of samples, we provide FID scores of DiffEnc-32-4 and VDMv-32 on CIFAR-10 in Table 8. We see from these, that the FID scores for the two models are similar and that it makes a big difference to the score whether we use the train or test set to calculate it and how many samples we use from the model. The scores are better when using more samples from the model and (as can be expected) better when calculating with respect to the train set that with respect to the test set.

## W    SUM HEATMAP OF ALL TIMESTEPS

A heatmap over the changes to $\mathbf{x}_t$ for all timesteps $t$ and all ten CIFAR-10 classes can be found in Fig. 6. Recall in the following that all pixel values are scaled to the range $(-1, 1)$ before they are given as input to the model. The DiffEnc model with a trainable encoder is initialized with $\mathbf{y}_\phi(\lambda_t) = 0$, that is, with no contribution from the trainable part. This means that if we had made this heatmap at initialization, the images would be blue where values in the channels are more than 0, red where values are less than 0 and white where values are zero. However, we see that after training, the encoder has a different behaviour around edges for $t < 0.8$. For example, there is a white line in the middle of the cat in the second row which is not subtracted from, probably to preserve this edge in the image, and there is an extra "outline" around the whole cat. We also see that for $t > 0.8$, the encoder gets a much more general behaviour. In the fourth row, we see that the encoder adds to the entire middle of the image including the white line on the horse, which would have been subtracted from, if it had had the same behaviour as at initialisation. Thus, we see that the encoder learns to do something different from how it was initialised, and what it learns is different for different timesteps.

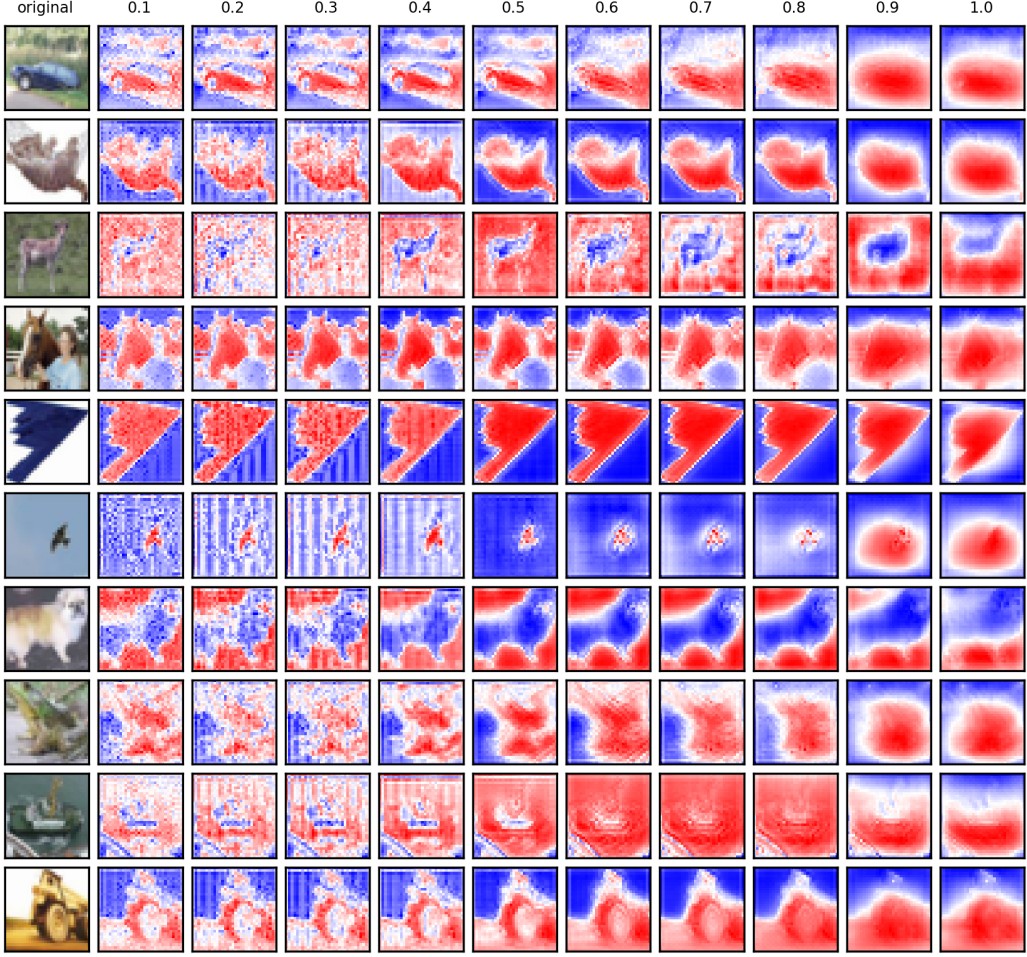

Figure 6: Change of encoded image over a range of depths: $(\mathbf{x}_t - \mathbf{x}_s)/(t - s)$ for $t = 0.1, ..., 1.0$ and $s = t - 0.1$. Changes have been summed over the channels with red and blue denoting positive and negative changes, respectively. For $t$ closer to 0 the changes are finer and seem to be enhancing high-contrast edges, but for $t \rightarrow 1$ they become more global.

