# OpenReview forum: "DiffEnc: Variational Diffusion with a Learned Encoder"
_ICLR.cc/2024/Conference — ICLR 2024 poster_

### Official Review · Reviewer_jJ4J · 2023-10-31

**Soundness:** 3 good
**Presentation:** 4 excellent
**Contribution:** 2 fair
**Rating:** 6
**Confidence:** 4

**Summary:**

The authors propose a version of diffusion models with a simple trainable encoder parameterized as a simple convex combination between  the data $\mathbf{x}$ and a neural network $f(\log SNR(t))$. parameterized by the signal to noise ratio ($SNR(t)$), a function of time.

**Strengths:**

- The paper is generally well-written and easy to read.
- The formulation for learnable encoders is simple and clear, and involves minimal added components from the original diffusion model.
- All derivations are well-documented in the appendix.

**Weaknesses:**

- The benefit of the encoder is unclear. There is limited exploration of the learned encoded space outside of the visualization provided in Figure 1. What is gained by this formulation?
- In spite of the added theoretical overhead incurred by the learned encoder, there is minimal empirical improvement over existing baselines (BPD 2.65 -> 2.63 on the CIFAR10 dataset).
- Empirical validation is slim. Model performance is only evaluated in terms of BPD on two simple datasets: MNIST and CIFAR10. MNIST is generally no longer used as a density estimation benchmark. As a result, I wonder is the performance generalizable? Is there improvement in terms of any other metrics? There is a lot of research on how likelihood can be a poor proxy for generative model performance. For example, a lot of improvements in BPD can be attributed to modeling semantically unimportant high frequency noise components [1, 2].
- Parameterization of the learned encoder is very simplistic. Why is the encoder conditioned only on the time $\lambda_t$ and not the data $\mathbf{x}$ itself? While I do appreciate elegant derivations, the justification for this parameterization (Appendix I) -- that this makes the math nice -- does not feel entirely sufficient.

[1] A note on the evaluation of generative models. https://arxiv.org/abs/1511.01844

[2] Soft truncation: A universal training technique of score-based diffusion model for high precision score estimation. https://arxiv.org/abs/2106.05527

**Questions:**

It is still not entirely clear to me what is learned by the encoder model. Can the authors directly visualize the output of the learned encoder model at different times $t$?

How does Section 7 (v-prediction parameterization) relate to the rest of the paper? Is it used for experiments in Section 8?

In which experiments in Section 8 were a fixed noise schedule used, and in which were a learned noise schedule used?

---

> ### Author Response · Authors · 2023-11-15
>
> Thank you very much for your review which helped up improve the article.
>
> > The benefit of the encoder is unclear.
>
> In a sense we are asking the question: “Can a diffusion model benefit from a non-trivial time-dependent encoder”. Since we achieve state of the art results on CIFAR-10, we say that the answer is “yes”. Since the encoded image is still in the full image space, the encoding is not directly applicable for representation learning. However, as we mention in the new section 7 (old section 9), it would be an interesting direction for future research to explore how distillations of the time-dependent encodings could be used for representation learning.
>
> > In spite of the added theoretical overhead incurred by the learned encoder, there is minimal empirical improvement over existing baselines
>
> Our large models achieve state of the art on CIFAR-10 showing a significant improvement over a model that uses the exact same architecture for the generative model. This is an extremely well-studied benchmark so it can be expected that it is difficult to improve SOTA with a lot. Furthermore, we continued training our models until 8 million steps and saw a further performance increase to 2.62 as seen in the updated Table 1. (The VDM paper uses 10 million steps.)
>
> > Empirical validation is slim
>
> We are currently training models on downscaled ImageNet 32x32. Due to the size of the models we do not have results ready for this response, however, we expect to have them on Monday, where we will share them in a new response.
>
> > Is there improvement in terms of any other metrics? There is a lot of research on how likelihood can be a poor proxy for generative model performance.
>
> As mentioned in your reference [1] likelihood and visual quality of samples are not directly linked and therefore the metric used to train a model is important when choosing the application. Since we show that our model can achieve good results on likelihood, and likelihood is important in the context of semi-supervised learning, it would be an interesting direction for future research to use this kind of model for classification in a semi-supervised setting. We have added this possible application to section 7. FID scores are a quite noisy metric, but we have added them to appendix R, and they show that FID scores for the VDM model with v parametrization and DiffEnc are not statistically different.
>
> > Parameterization of the learned encoder is very simplistic. Why is the encoder conditioned only on the time?
>
> The encoder is dependent on both the time and the original image. As we wrote in the previous version of the article in section 3: $x_t = f_{\phi}(x, t)$. We have made this more clear in the beginning of the section. In our new figure 1, we also visualize the setup and show how the encoder takes both an image and time as inputs.
>
> > ...the justification for this parameterization (Appendix I) -- that this makes the math nice -- does not feel entirely sufficient.
>
> The reason for using $\alpha_t$ multiplied with the data in the encoder is that we want the encoder to be the identity for t = 0 and to send the mean to 0 for t = 1. This is important for minimizing the reconstruction and latent losses as we mention in the beginning of the new section 4 (old section 6). The argument about making the math nicer is only to justify that instead of using just $\alpha_t$, we use $\alpha_t^2$. We have rewritten appendix I to make this clearer.
>
> > Can the authors directly visualize the output of the learned encoder model at different times
>
> Good idea. In our new figure 1, we have some examples of the encoded images. In general, the encoder seems to enhance edges for lower t and bring the mean of the image closer to 0 for higher t, however while keeping more signal for higher t than the non-trainable encoder would. We have clarified this in results in the new section 5 (old section 8.2).
>
> > How does Section 7 (v-prediction parameterization) relate to the rest of the paper?
>
> As we write in the new section 4 (old section 7), we use the v parametrization in our experiments. The exact equations we use for our loss are using the new numbers (19) and (20), the old numbers were (38) and (39).
>
> > In which experiments in Section 8 were a fixed noise schedule used, and in which were a learned noise schedule used?
>
> Thank you for bringing this lack of clarity to our attention. As we mention in the experimental setup of the new section 5 (old section 8.1), we used fixed endpoints for the noise schedule for the large models and experimented with both fixed and learned endpoints for the small models. Table 2 has results of the large models, so these are with a fixed noise schedule, and we have added this to the caption. In table 3 and 4, we write in the column “noise” which results are for a fixed noise schedule and which are for a trainable schedule. In the caption of table 5, we write that these results are for a fixed noise schedule.

---

### Official Review · Reviewer_Hg56 · 2023-10-31

**Soundness:** 3 good
**Presentation:** 3 good
**Contribution:** 3 good
**Rating:** 6
**Confidence:** 4

**Summary:**

The paper extends variational diffusion models (VDM) with a time dependent encoder. Instead directly noising the input data points (here images), they are first encoded and then noised. There are two encoding schemes introduced. In one, the encoder has no trainable parameters and is simply a rescaling of the input. In the other, the encoded images are a reweighting of the noise-free image and the output of a time-dependent image-to-image neural network. The authors derive the corresponding loss functions and analyze the model in its infinite depth limit. The results show that the diffusion model equipped with a new encoder achieves state-of-the-art BPD on CIFAR-10.

**Strengths:**

The high-level idea for the paper is quite natural and something that somebody was bound to try because of its potential impact. Overall, I found the writing fairly clear, with some exceptions that I will mention in the next section. The analysis of the method is fairly extensive and supported by lot of details in the appendix, though these are primarily mathematical proofs and not necessarily an exploration of design decisions that have a high level of practical significance.

**Weaknesses:**

The presentation of the method, namely section 3 and 6 could be improved significantly. There are a lot of variable names, and I had to read through the section many time in order to understand what was happening, even though the final procedure is not that complex. Moving the figure provided in Appendix A to the main text might be helpful in this regard. Or you could include an algorithm, or simply a link to your code, as these would all be easier to parse as someone familiar with common diffusion code and parlance. The presentation is also strong emulating the style of VDM even though the method has interesting consequences and potential applications on diffusion models more broadly, and simplifying the description of the algorithm to rely less on graphical models and equations and simply describe where noising and denoising are happening might make the approach more accessible to a broader audience.

Beyond the presentation, the results are pretty limited. Given that the proposed method is an incremental change to VDM, the performance improvement is not overwhelming. It feels like an improvement on par with what might be possible through better hyperparameter tuning or by increasing the model capacity (which is implicitly part of what this method is doing, at least in the case where the encoder has learnable parameters). The results being limited to CIFAR-10 compound my intuition that the results aren't very notable, because it's not clear if they generalize. Though I think method appears sound and significant and general results seem possible, the current results feel like a negligible improvement on a relatively small dataset. It would be great to see results paralleling the original VDM model. If the improvements spanned all the datasets considered in the original paper, I would be more convinced that the proposed encoder is doing something significant.

**Questions:**

Do you have a sense for what the weaknesses of your gradient approximation (Section 6) are? It seems that the approximation will be very bad for t near 1. Does this effect the learning of the encoder in a serious way? It might be helpful to explore this a bit more if it is one of the more impactful approximations required to your method practical.

What is the intended take-away from Figure 1? Is the significance simply that the learned encoder captures the spatial frequencies that we might expect at each time in the diffusion process? Or do you also conclude that there is an implicit form of segmentation happening? The later feels potentially quite profound but needing more justification while the former feels potentially unsurprising, especially if the gradient signal is just incredibly noisy. It could just be learning very little for t near 1.0.

---

> ### Author Response · Authors · 2023-11-15
>
> Thank you very much for your review. It has been very helpful for making revisions.
> > The presentation of the method, namely section 3 and 6 could be improved significantly.
>
> Thank you for this valuable feedback. We have made a thorough rewrite of the article and feel that this makes the presentation of the method easier to follow.
>
> > Moving the figure provided in Appendix A to the main text might be helpful...
>
> Thank you very much for this suggestion. We have made a visualization of how DiffEnc works and have added it as our figure 1. We feel this clarifies the overall idea.
>
> > ...the performance improvement is not overwhelming...
>
> The result of the large models is new state of the art on CIFAR-10 showing a significant improvement over a model that uses the exact same architecture for the generative model. This is an extremely well-studied benchmark so it can be expected that it is difficult to improve SOTA by a substantial amount. Furthermore, we continued training our models until 8 million steps and saw a further performance increase to 2.62 as seen in the updated Table 1. (The VDM paper uses 10 million steps.)
>
> > ... an improvement on par with what might be possible through better hyperparameter tuning…
>
> Since we train a baseline VDM using V parametrization, which uses the exact same hyperparameters as the DiffEnc model, if better results could be gained for the VDM with better hyperparameter tuning, then it would likely be possible to get the same improvements for DiffEnc.
>
> > ... or by increasing the model capacity (which is implicitly part of what this method is doing, at least in the case where the encoder has learnable parameters).
>
> The encoder is not used during generation. Thus, our work shows that using an encoder enables us to find a better generative model without increasing the number of parameters of the generative model.
>
>
> > The results being limited to CIFAR-10...
>
> We are currently training models on downscaled ImageNet 32x32. Due to the size of the models we do not have results ready for this response, however, we expect to have them on Monday, where we will share them in a new response.
>
> > Do you have a sense for what the weaknesses of your gradient approximation (Section 6) are? It seems that the approximation will be very bad for t near 1.
>
> The approximation of the gradient will be bad for the trainable encoder if $y_{\phi}(\lambda_t) - dy_{\phi}(\lambda_t)/dt$ is large. One might think that this would force $y_{\phi}(\lambda_t)$ to be close to $dy_{\phi}(\lambda_t)/dt$, however inspection of some examples show that this is not the case. The reason might be that the normed part of the loss is scaled with $\alpha_t^2$, and therefore the difference $(y_{\phi}(\lambda_t) - dy_{\phi}(\lambda_t)/dt)$ can be rather large without affecting the total loss notably. Of course, one might get even better results with a better approximation, and this would be an interesting direction for future research.
>
> > What is the intended take-away from Figure 1?
>
> (In the new version of the paper this is figure 2) In short, the take-away is that the encoder learns to do something different from how it was initialised, and what it learns is different for different timesteps. We have reworded the figure caption and added a sentence in the results of the new section 5 (old section 8.2) to clarify. We did not have room for the details in the paper, but we have added them to appendix S.

---

> ### Comment · Reviewer_Hg56 · 2023-11-22
>
> Thank you for your response and your changes to the submission. I think these changes have substantially improved the clarity and credibility of the work, and I will be raising my score to a 6.

---

### Official Review · Reviewer_neQL · 2023-11-04

**Soundness:** 2 fair
**Presentation:** 2 fair
**Contribution:** 3 good
**Rating:** 6
**Confidence:** 1

**Summary:**

This work delves into the realm of diffusion models through the lens of hierarchical Variational Autoencoders (VAEs). The authors put forth a novel iteration of diffusion models that preserves the advantages of VAEs. To be more specific, they introduce a time-dependent DiffEnc, resulting in encoder variations across different time steps. Furthermore, the authors conduct an in-depth exploration of the assumption that forward and backward variances are equal, underpinned by thorough theoretical analysis. In addition, empirical findings are presented to substantiate the efficacy of the proposed DiffEnc.

**Strengths:**

I believe that implementing a trainable encoder within the context of the diffusion model represents a promising avenue for enhancing diffusion models, particularly in terms of ELBO optimization. This paper offers comprehensive insights into the derivation and analysis, rendering it accessible and straightforward to grasp. The experimental findings are not only persuasive but also harmonize effectively with the theoretical framework. For instance, in Figure 1, we observe a logical outcome indicating that during the initial stages, the encoder primarily attends to local patterns, such as edges. Simultaneously, as the value of $t$ increases, it places greater emphasis on capturing global structures.

**Weaknesses:**

1. The organization of sections is perplexing. It's challenging for me to discern whether Section 2 serves as an introductory section or is meant to highlight one of your contributions.
2. The absence of a central theorem throughout the paper poses a difficulty for readers in anticipating the direction of the derivations and what to expect.

**Questions:**

I don't have specific questions regarding the main text, but it's clear that there's a need for improvements in the organization of the content.

---

> ### Author Response · Authors · 2023-11-15
>
> Thank you very much for your review, especially your helpful suggestion of clarifying the main equation.
>
> > It's challenging for me to discern whether Section 2 serves as an introductory section or is meant to highlight one of your contributions.
>
> Good point. Section 2 is meant as an introductory section, to introduce the notation used for variational diffusion models. We have added a sentence at the beginning of the section to clarify this and changed the title to “Preliminaries on Variational Diffusion Models”.
>
> > The absence of a central theorem throughout the paper poses a difficulty for readers in anticipating the direction of the derivations
>
> Thank you for this suggestion. We have added in the introduction which equation is the main expression of the DiffEnc loss - Equation (18) - and sketched how to arrive at it.
>
> > it's clear that there's a need for improvements in the organization of the content.
>
> We have taken your comment to heart and made a thorough restructuring of the sections. We think that this made the organization more intuitive by making the separation between DiffEnc and the standard diffusion approach clearer.

---

### Official Review · Reviewer_g9hi · 2023-11-07

**Soundness:** 3 good
**Presentation:** 3 good
**Contribution:** 2 fair
**Rating:** 5
**Confidence:** 2

**Summary:**

The paper proposes to model the integration of a learnable data and time-dependent mean function for diffusion models. The paper investigates the parametrisation of the diffusion losses and shows optimal variance schedules for the forward and reverse processes. The paper shows density estimation results on CIFAR 10 and MNIST and achieves the best performance compared to relevant baselines.

**Strengths:**

The paper is well written and easy to follow. It adds a simple yet interesting addition to diffusion by introducing a mean shift to the forward diffusion while not being required in the sampling process and therefore ensuring its scalability. The theoretical analysis of the different noise variances adds an interesting flavour too. The results seem to indicate that the added encoder improves the performance in terms of bits per dimension.

**Weaknesses:**

The paper has very limited evaluation and doesn't compare to some of relevant baselines that are even mentioned in the paper, like latent diffusion and only compares on Cifar-10 and MNIST. Furthermore, it mentions that some methods only show improvement after longer training, hinting at potential inconsistencies in the results in case of slightly different training setups due to not training till convergence. It is hard to judge whether the proposed changes are a significant improvement due to this.
The ablation studies of the different methods are interesting but it's unclear what insight is gained from this, e.g. what does it mean that the loss is dominated by the diffusion loss?
Lastly, the theoretical analysis of the variance ratio is interesting but unclear how it fits into the bigger story of the paper.

**Questions:**

- In section 3, I first missed the added subscript in $x_t$ in eq 13. It might make it easier to read to change the notation from $x_t$ to e.g. $e_t$ or something similar to be clear there.
- in section 6, I was not fully following the parametrisation of $x(\lambda_t)$ - how does this related to eq (13)? E.g. does eq. (29) indicate that $q(z_t|x) = N(a_t^3 x; ...)$?

---

> ### Author Response · Authors · 2023-11-15
>
> Thank you very much for your review. In the new version of the article, we have done our best to address your concerns.
> > The paper has very limited evaluation...
>
> We are currently training models on ImageNet 32x32. However, SOTA models are large and require access to extensive compute. We will share results later in the rebuttal period.
>
> > ...doesn't compare to some of relevant baselines that are even mentioned in the paper, like latent diffusion...
>
> In Table 1, we compare to all state of the art likelihood-based models for CIFAR-10 without data augmentation. Since we did not use data augmentation on CIFAR-10, we have only compared with other results on non-augmented CIFAR-10. In the latent diffusion paper by Rombach et al 2022 “High-Resolution Image Synthesis with Latent Diffusion Models”, they do not report any likelihoods on CIFAR-10, therefore they have not been included in the table.
>
> > …it mentions that some methods only show improvement after longer training
>
> By design, the DiffEnc and VDMv models have the same initial loss. However, the trends differ, especially from around 2 million training steps. Which means that it is only after more than 2 million steps the mean losses differ significantly. We would not say that this shows any kind of inconsistency in the results. We have made this clearer in the description of the results in table 5 which was in the old section 8.2. However, we have had to move table 5 to appendix Q due to the page restriction.
>
> > ...what does it mean that the loss is dominated by the diffusion loss?
>
> By construction, the VDM model has very small latent and reconstruction loss when using the fixed signal to noise ratio endpoints from the original article. Thus, the diffusion loss is where there is the biggest potential for improvement. We have added a mention of this in results in the new section 5 (old section 8.2) when discussing tables 3-5.
>
> > In section 3, I first missed the added subscript in x_t in eq 13
>
> Thank you for pointing us towards this point of potential confusion. We use $x_t =x_{\phi}(\lambda_t)$, to remind the reader that the encoded image is dependent on both x and t through $\lambda_t$. We think that changing the notation to not use x would obfuscate this dependency. We have added “dependent on x and t” when we introduce the encoder in section 3 to clarify.
>
> > In section 6, I was not fully following the parametrisation of $x(\lambda_t)$ how does this related to eq (13)?
>
> From the old section 5 and onwards we write $x_t = x_{\phi}(\lambda_t)$ to include the dependence of the encoder on the learned parameters, $\phi$, and the log of the signal to noise ratio, $\lambda_t$. In the new version we introduce  $x_{\phi}(\lambda_t)$ from the beginning of section 3, hopefully, this makes it clearer. When given x, we have for the non-trainable encoder $z_t = \alpha_t^3x$ and for the trainable encoder $z_t = \alpha_t(\alpha_t^2x + \sigma_t^2 y_{\phi}(\lambda_t))$ where $y_{\phi}$ is the trainable part of the encoder.

---

### Author Response · Authors · 2023-11-15

We thank all reviewers for their helpful feedback. We have taken all of it into account and we believe that it has substantially improved the paper.
We have replied to the specific comments and questions of each reviewer individually. Here we will focus on points raised by more than one reviewer.

Generally, reviewers g9hi, neQL, and Hg56 comment positively on the main contribution.

Further, reviewers g9hi, Hg56, and jJ4J remark that the writing of the paper is generally clear and easy to follow.

neQL, Hg56, and jJ4J also comment favourably on the thorough derivations supplied for the theoretical foundation of the method.

Below, we provide a summary of the general changes and additions we have made to our submission to address the concerns of the reviewers:

- We have added a new figure 1 which illustrates the overall idea of our DiffEnc model and how it differs from the standard diffusion approach.

- We acknowledge that the paper is equation-heavy and have therefore clarified in the introduction which equation contains the main result for the DiffEnc loss and how we will arrive at this expression.

- We have restructured the sections to make the story more coherent, especially in response to reviewer neQL and Hg56. The old sections 2 and 4 are now section 2 “Preliminaries on Variational Diffusion Models”. The old sections 3 and 5 are now section 3 “DiffEnc”. In addition, the most important parts of the old section 7 have been combined with the old section 6 to make a new section 4 “Parameterization of the Encoder and Generative Model”.

- We have added a new figure (figure 3), which compares some samples of the models to illustrate the visual quality of the samples.

- We have simplified our notation especially in the new section 4, the old section 6.

- In response to the demand for more evaluation, we note that our current results are all run with at least three seeds (as opposed to common one seed industry practice) and that we have begun training models on ImageNet 32x32. Due to the size of the models we do not have results ready for this response, however, we expect to have them before the rebuttal period ends (on Monday), where we will share them in a new response.

---

### Author Response · Authors · 2023-11-20
**Update After Running Experiments on ImageNet 32x32 for 1 Million Steps**

To meet the need for more evaluation, we have trained both a VDM with v-parametrization and a DiffEnc model with trainable encoder for 1 million steps on the ImageNet 32x32 dataset from Chrabaszcz et al. (2017) "A downsampled variant of imagenet as an alternative to the cifar datasets" from the official ImageNet website: https://image-net.org/download-images.php. We have added these results to our table 1. We have also added an extra comparison in the table (Lipman2023), since the results on ImageNet 32x32 from the comparisons we had before were all on the Van den Oord et al. (2016) "Pixel recurrent neural networks" version of the dataset, which is no longer officially available.

We get a result of 3.52 BPD for both the DiffEnc and VDM with v-parametrization which is state-of-the-art to the best of our knowledge.

For the camera-ready version, we will train two more seeds, so that we have three seeds like on CIFAR-10 and we will train for longer, since 1 million steps is very few compared to how many steps are usually used for ImageNet.

---

### Comment · Area_Chair_tAfd · 2023-11-20
**To all reviewers: Please respond to the authors' rebuttal**

Dear reviewers,

The window for interacting with authors on their rebuttal is closing on Wednesday (Nov 21st). Please respond to the authors' rebuttal as soon as possible, so that you can discuss any agreements or disagreements. Please acknowledge that you have read the authors' comments, and explain why their rebuttal does or does not change your opinion and score.

Many thanks,

Your AC

---

### Meta-Review · Area_Chair_tAfd · 2023-12-08

**Metareview:**

This paper received mixed review scores. The authors addressed several concerns of the reviewers during the rebuttal and discussion phase, improving the clarity of the writing and adding additional experiments on Imagenet32x32. During the discussion phase, two out of four reviewers have increased their score, changing from marginally below the threshold to marginally above the acceptance threshold. The reviewer whose recommendation is that the manuscript is marginally below the threshold has not engaged with the authors during the rebuttal period, nor during the follow up discussion period between reviewers and AC. After reading the paper and the rebuttal, a considerable part of their concerns seems to be addressed with the rebuttal and the revisions. Given the more positively leaning opinion from the other reviewers, and after reading the paper, it is the perspective of the AC that the ideas presented in this submission are clearly presented, and that although the experimental validation does not lead to ground breaking improvements, the paper brings value to the community. Therefore, the recommendation is to accept this paper.

**Justification For Why Not Higher Score:**

The empirical improvements are quite minimal.

**Justification For Why Not Lower Score:**

The only reviewer who kept their vote for rejecting this paper after the rebuttal phase has been unresponsive since posting their review, despite repeated requests for engagement. Based on reading the paper and the rebuttal, their concerns that seem to have been largely addressed. Given also the other positively leaning reviews, I vote for accepting this paper.

---

### Decision · Program_Chairs · 2024-01-16

Accept (poster)